# Text summarization via global structure awareness

**Jiaquan Zhang[1], Chaoning Zhang[1,\*], Shuxu Chen[2], Yibei Liu[1], Chenghao Li[1], Qigan Sun[1]**
**Shuai Yuan[1], Fachrina Dewi Puspitasari[1], Dongshen Han[1], Guoqing Wang[1], Sung-Ho Bae[2], Yang Yang[1]**

[1]University of Electronic Science and Technology of China
[2]Kyung Hee University

## Abstract

With the explosive growth of information, the volume of long documents has surged, and the cost of processing them continues to rise, making text summarization increasingly important. Existing studies primarily focus on model enhancements and sentence-level pruning based on contextual dependencies and semantic patterns. Although some approaches leverage large language models (LLMs) for text summarization and achieve higher accuracy, they incur substantial computational costs and often overlook global structural modeling. Consequently, summarized texts may lose critical logical chains, disrupting coherence and weakening downstream task performance. To address these issues, we propose GloSA-Sum, a novel text summarization framework that performs global structural analysis of texts via topological data analysis (TDA), enabling efficient summarization while preserving semantic cores and logical dependencies. Specifically, we construct a semantic-weighted graph from sentence embeddings, where persistent homology identifies core semantics and logical structures, preserved in a "protection pool" as the backbone for summarization. We design a topology-guided iterative strategy, where lightweight proxy metrics approximate sentence importance to avoid repeated high-cost computations, thus preserving structural integrity while improving efficiency. To further enhance long-text processing, we propose a hierarchical strategy that integrates segment-level and global summarization. Experiments on multiple datasets demonstrate that GloSA-sum reduces redundancy while preserving semantic and logical integrity, striking a balance between accuracy and efficiency, and further benefits LLM downstream tasks by shortening contexts while retaining essential reasoning chains.

## 1 Introduction

With the explosive growth of digital information Zheng et al. (2025b;a), human society generates massive volumes of long documents daily. Directly processing such lengthy texts creates efficiency and accuracy bottlenecks for downstream NLP tasks. In particular, when LLMs Zheng et al. (2026a;b) are required to handle long contexts, redundant information quickly exhausts the context window, increases computational cost, and distracts the model from focusing on core content. Therefore, effectively summarizing long texts without losing key information and logical chains has become a central challenge in NLP.

Existing methods aim to improve summarization performance from different perspectives. A straightforward and widely adopted approach is sentence-level pruning based on local semantic similarity or statistical features. A typical example models Mihalcea & Tarau (2004); Erkan & Radev (2004) a document as a graph of sentence similarities and then applies graph ranking to

---

*∗ Corresponding Author. Contact: `chaoningzhang1990@gmail.com`.
Additional contacts: `jiaquanzhang2005@gmail.com`, `ccccsx322@gmail.com`,
`2023091204024@std.uestc.edu.cn`, `2024090909011@std.uestc.edu.cn`,
`lch17692405449@gmail.com`, `sunqigan0206@gmail.com`, `puspitasari-dewi@outlook.com`,
`gqwang0420@uestc.edu.cn`, `han-0129@khu.ac.kr`, `shbae@khu.ac.kr`,
`yang.yang@uestc.edu.cn`

select salient sentences. This idea is also extended to multi-document summarization by incorporating document-level weights to better capture cross-document importance Wan (2008). Other works formulate sentence selection as an optimization problem to balance content coverage and redundancy Clarke & Lapata (2008); Lin & Bilmes (2011). These methods are efficient and intuitive, yet their reliance on local similarity or shallow features often limits their ability to capture global semantic structures and long-range logical dependencies. Some studies focus on improving model architectures to enhance summarization. BERTSum Liu & Lapata (2019) employs BERT-based encoders with sentence-level classifiers to strengthen contextual representations, while MatchSum Zhong et al. (2020) reformulates the task as a candidate–summary–document matching problem to ensure holistic consistency. TexShape Kale et al. (2024) combines pretrained language models with a neural module to construct information-theoretic sentence embeddings based on mutual information, enabling controllable summarization while preserving useful content and filtering sensitive information. Although these approaches improve representational power and accuracy, they often face scalability and efficiency bottlenecks when applied to very long documents. Another line of research leverages LLMs for summarization Zhang et al. (2024); Azher et al. (2024). Despite their strong performance, LLM-based methods incur substantial inference costs and computational overhead, which limit their applicability in large-scale long-text scenarios.

To address these issues, we explore whether it is possible to reduce excessive resource consumption while maintaining summarization quality. TDA Uchendu & Le (2024); Wasserman (2018) provides a global perspective for capturing semantic structures and logical dependencies, enabling the extraction of key reasoning chains and thus preserving the overall skeleton of a document during summarization. Based on this insight, we propose a Global Structure-Aware TDA-based Summarization Framework (GloSA-sum). Specifically, we encode sentences into high-dimensional semantic embeddings and construct a weighted undirected graph where edge weights jointly reflect semantic similarity and positional distance, thereby balancing global semantic relations with local discourse coherence. We then apply persistent homology to track the birth and death of semantic structures across scales, distinguishing short-lived noise from persistent structural features. Zero-dimensional homology (H0) corresponds to semantic clusters that reveal the document's core themes, while one-dimensional homology (H1) captures loop structures that reflect cross-paragraph logical dependencies. By selecting persistent topological features, we extract a robust semantic and logical backbone that is preserved throughout the summarization process. However, directly applying TDA to long-text summarization introduces efficiency challenges: repeatedly computing persistent homology during iterative summarization is computationally prohibitive. Consequently, we propose a Protected Pool mechanism, which performs a one-time topological analysis to identify and fix the document's semantic and logical backbone. Persistent homology is computed only once, while subsequent iterations rely on lightweight proxy metrics to evaluate and filter non-critical sentences, achieving efficient iterative summarization. Furthermore, to handle ultra-long texts, we design a hierarchical summarization strategy. The document is first partitioned into paragraphs or semantic segments, within which local TDA-based summarization is executed in parallel; the locally summarized results are then globally integrated and refined with a lightweight topological constraint to ensure cross-paragraph logical consistency.

The contributions of this work are as follows:

- We introduce TDA into text summarization for the first time, offering a novel global structural perspective that explicitly models and preserves both semantic clusters and cross-paragraph logical dependencies.

- We propose a one-time topological analysis and proxy-based iterative summarization strategy, where the Protected Pool mechanism avoids repeated persistent homology computations and achieves an effective balance between efficiency and semantic integrity.

- We design a hierarchical summarization framework that enables coordinated local summarization and global integration, thereby significantly enhancing scalability and robustness in long-text scenarios.

- Extensive experiments show that GloSA-sum outperforms strong baselines in summarization while also enhancing LLM downstream tasks by reducing context length and preserving essential reasoning chains.

## 2 RELATED WORK

### 2.1 TEXT SUMMARIZATION METHOD

Text summarization methods can be broadly categorized into two lines: model-level improvements and sentence-level pruning, which exploit contextual dependencies and semantic patterns. Early text compression and summarization approaches primarily relied on sentence-level pruning, treating sentences as atomic units to be selected or discarded based on estimated importance. Graph-based methods such as TextRank Mihalcea & Tarau (2004) and LexRank Erkan & Radev (2004) construct sentence similarity graphs and apply centrality-based ranking to identify core sentences, demonstrating the effectiveness of unsupervised approaches across diverse corpora, but offering limited modeling of global discourse structure and cross-paragraph dependencies. To alleviate these limitations, integer linear programming frameworks Clarke & Lapata (2008) formulate sentence selection as an optimization problem that balances content coverage and redundancy. More recent unsupervised methods further extend sentence-level pruning. RankSum Joshi et al. (2022) fuses multiple ranking signals to estimate sentence importance without annotated data, while Jie et al. (2024) introduces a differentiable knapsack module to enable learnable length control in extractive summarization. Despite their efficiency and simplicity, sentence-level pruning methods often fail to capture fine-grained semantic coherence and global logical flow due to their coarse granularity. From the model-level perspective, Gu et al. (2022) employs memory networks to iteratively track selected content and reduce redundancy, and Goyal et al. (2019) combines recurrent neural networks with arithmetic coding for lossless semantic compression. With the emergence of large language models, recent work explores their potential for lossless semantic compression, including fine-tuning LLMs toward the Shannon limit Mittu et al. (2024) and leveraging next-token predictive distributions for entropy coding Mao et al. (2025).

### 2.2 TOPOLOGICAL DATA ANALYSIS IN NLP

TDA offers a systematic framework to capture global structures in high-dimensional and complex data. Wu et al. (2022) integrates persistent homology features over embeddings into deep learning models to detect multiple types of textual contradictions. Their results show that topological features significantly outperform standard baselines in contradiction detection. Recent research further extends TDA to interpretability and discourse-level analysis. For example, Proskurina et al. (2023) applies TDA to linguistic acceptability judgments. By designing new topological features such as chordality and matching number on attention graphs, they achieve performance gains over fine-tuning baselines in both English and Russian, and further reveal correspondences between specific attention heads and linguistic phenomena. In parallel, Jain et al. (2024) investigates discourse coherence through TDA in Beyond Words and shows that topological signatures capture semantic flow and logical structure within documents. Despite these advances, existing works focus primarily on interpretability or classification tasks, while the systematic integration of TDA into large-scale text compression and summarization frameworks remains underexplored. Our work addresses this gap by leveraging persistent homology Edelsbrunner et al. (2008) to identify and preserve robust topological features for summarization, thereby ensuring both semantic coherence and logical consistency in the output.

## 3 METHODOLOGY

### 3.1 PRELIMINARY KNOWLEDGE OF TDA AND HOMOLOGY GROUPS

TDA is a framework originating from algebraic topology that captures the intrinsic structural patterns of complex data. It represents a dataset as a collection of points in a high-dimensional space (a point cloud) and examines how these points are connected. To extract meaningful structures, TDA employs persistent homology, which tracks how topological features emerge and disappear as the observation scale changes. Intuitively, this process is akin to gradually increasing the resolution for observing data, much like continuously zooming in and out. Features that vanish quickly are usually seen as noise, while those that persist across a wide range of scales are regarded as meaningful and robust patterns. Through this multi-scale view, TDA can reveal fundamental elements of structure, such as connected clusters, loops, and higher-dimensional cavities that are otherwise difficult to cap-

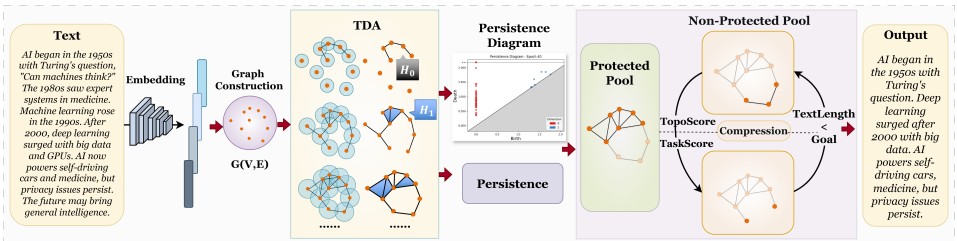

Figure 1: Overall of GloSA-sum

ture. Formally, these topological features are characterized by homology groups, which summarize the structure of a simplicial complex at different dimensions. In essence, homology groups provide the algebraic foundation of TDA by formally characterizing topological features such as connected components and cycles, while TDA operationalizes these concepts through persistent homology to analyze complex datasets across multiple scales. The $k$-th homology group is defined as

$$H_k(K) = \frac{\ker \partial_k}{\operatorname{im} \partial_{k+1}},\tag{1}$$

where $\partial_k$ denotes the boundary operator mapping $k$-chains to $(k-1)$-chains, and $H_k$ intuitively captures $k$-dimensional "holes" that cannot be expressed as the boundary of higher-dimensional objects. In particular, $H_0$ (zero-dimensional homology) corresponds to connected components, which in the context of text analysis can be interpreted as core semantic themes or independent clusters of meaning that form the backbone of the discourse. Meanwhile, $H_1$ (one-dimensional homology) corresponds to non-trivial loops or cycles, which often manifest in text as logical loops or recurrent argumentative structures that link different parts of the document. Although higher-order homology groups such as $H_2$ describe voids or cavities in data, They are less relevant for sequence-like data such as text. Therefore, in this work, we primarily focus on $H_0$ and $H_1$, as they directly correspond to the preservation of semantic themes and logical structures, which are crucial for maintaining coherence in compressed text representations.

## 3.2 Overview of GloSA-sum

As shown in Figure 1, GloSA-sum is a global structure-aware summarization framework. It performs a one-time persistent homology analysis to extract core semantic clusters (H0) and logical cycles (H1), preserved in a Protected Pool as the document backbone. Guided by lightweight proxy metrics and a hierarchical compression strategy, the method progressively removes redundancy while ensuring semantic fidelity and logical consistency for long texts.

## 3.3 Semantic graph construction

To enable the application of TDA, we first transform the input document $D$ into a representation suitable for topological analysis. Each sentence is then encoded using a pretrained sentence encoder into a semantic embedding $\mathbf{e}_i$. To eliminate scale discrepancies across sentences, we apply normalization to all $\mathbf{e}_i$, enabling stable similarity computations. Based on the sentence embeddings, we construct a weighted undirected graph:

$$G = (V, E)\tag{2}$$

where the node set $V$ corresponds to the sentence set $S$, and the edge set $E$ encodes the semantic and sequential relations between sentences. To capture global semantic relations while maintaining graph sparsity, we adopt a mutual $k$-nearest neighbor strategy Baoli et al. (2004) with an adaptively determined neighborhood size. The value of $k$ grows logarithmically with the document length, ensuring that shorter texts are not over-connected while longer texts preserve sufficient structural connectivity. An undirected edge $(i, j) \in E$ is established if and only if sentence $s_i$ lies within the adaptive neighborhood of $s_j$ and vice versa. Each edge is further assigned a hybrid weight $w_{ij}$:

$$w_{ij} = \alpha \cdot d_{ij}^{\text{sem}} + (1 - \alpha) \cdot \exp\left(-\frac{|i - j|}{\tau}\right)\tag{3}$$

where $d_{ij}^{\text{sem}} = 1 - \cos(\mathbf{e}_i, \mathbf{e}_j)$ denotes the semantic distance derived from cosine similarity between sentence embeddings, and $|i - j|$ represents the absolute positional distance between two sentences in the original sequence. The coefficient $\alpha \in [0, 1]$ serves as a fusion parameter that balances the contributions of semantic similarity and sequential adjacency. At the same time, the temporal decay factor $\tau$ controls the sensitivity of the sequential proximity term, ensuring that adjacent sentences exert a stronger influence than distant ones. The $w_{ij}$ scheme encodes both semantic proximity and sequential coherence, thus preserving argumentative continuity. Notably, the semantic distance $d_{ij}^{\text{sem}}$ is also retained as the primary metric for subsequent TDA.

### 3.4 PROTECTED POOL INITIALIZATION FROM TOPOLOGICAL ANALYSIS

Unlike prior iterative graph-based summarization methods, GloSA-sum performs persistent homology computation only once at the beginning, to identify the document's semantic and logical backbone. This one-time topological analysis permanently fixes the global structure, thereby avoiding repeated high-cost TDA computations and ensuring scalability to long documents. Concretely, we compute persistent homology over the point cloud $\mathbf{e}_1, \ldots, \mathbf{e}_n$, where each sentence embedding is treated as a discrete point in a high-dimensional semantic space. To approximate the underlying simplicial complex efficiently, we employ the Lazy Witness Complex Arafat et al. (2019) with a fixed proportion of landmark points. Persistent homology is computed up to dimension one, yielding persistence diagrams $D(0)$ and $D(1)$ corresponding to $H_0$ and $H_1$, respectively. Each topological feature, whether a connected component or a cycle, is quantified by its persistence length:

$$\ell = d - b, \tag{4}$$

where $b$ and $d$ denote the birth and death scales within the filtration. Then, we initialize the Protected Pool $\mathcal{P}$ to preserve the document's essential structural backbone permanently. The Protected Pool consists of two complementary components derived from different homological dimensions:

- Core themes ($H_0$): We select the top-$K$ longest-living $H_0$ features (i.e., the connected components with the greatest persistence), and collect the sentences associated with their landmark points into $\mathcal{P}_{H0}$. This guarantees that the primary semantic clusters of the document are retained while keeping the pool size controllable.

- Critical logical cycles ($H_1$): We select the top-$M$ most persistent $H_1$ cycles and aggregate all sentences participating in these cycles into $\mathcal{P}_{H1}$. This ensures that essential logical dependencies and discourse-level cycles are preserved.

Finally, the Protected Pool is defined as:

$$\mathcal{P} = \mathcal{P}_{H0} \cup \mathcal{P}_{H1}, \tag{5}$$

which jointly safeguards the document's semantic themes and logical structures throughout the compression process.

### 3.5 TOPOLOGY-GUIDED ITERATIVE COMPRESSION

Since the Protected Pool has already secured the global semantic backbone through a single persistent homology analysis, the subsequent compression process no longer requires recomputing TDA. Instead, redundant sentences are progressively removed using lightweight proxy metrics that approximate topological importance. This design preserves the global semantic and logical structure identified by the Protected Pool while avoiding the prohibitive cost of repeated persistent homology computations.

Specifically, at each iteration, every sentence $s_i \in S \setminus \mathcal{P}$ outside the Protected Pool is assigned a composite deletion priority score that jointly considers topological connectivity and task relevance. Sentences with lower scores are deleted earlier, ensuring that structurally important sentences remain protected. This scoring mechanism balances structural preservation with adaptability to downstream queries, making the compression process both computationally efficient and faithful to the global semantic structure.

$$\text{Score}(s_i) = \lambda \cdot \text{TopoScore}(s_i) + (1 - \lambda) \cdot \text{TaskScore}(s_i), \tag{6}$$

where $\lambda \in [0, 1]$ controls the relative weight of the two components.

**TopoScore Computation.** The topological proxy score $\text{TopoScore}(s_i)$ quantifies the structural importance of a sentence with respect to the semantic backbone captured by the Protected Pool. We adopt Dijkstra's algorithm to obtain the shortest-path length $\text{SPL}(s_i, s_j)$ from node $s_i$ to each protected node $s_j \in \mathcal{P}$. The semantic graph $G$ is constructed via a $k$-nearest-neighbor strategy, resulting in a sparse topology that allows fast shortest-path calculations even for long documents. The score is computed as:

$$\text{TopoScore}(s_i) = -\sum_{s_j \in \mathcal{P}} \text{SPL}(s_i, s_j), \tag{7}$$

Because of the negative sign, $\text{TopoScore}(s_i)$ values closer to zero indicate stronger connectivity to the structural skeleton (thus higher importance), whereas more negative values indicate weaker connectivity. For nodes that are not connected to any protected node (i.e., $\text{SPL}(s_i, s_j) = \infty$ for all $s_i$), we assign a large negative penalty to $\text{TopoScore}(s_i)$. This ensures that semantically peripheral and structurally isolated sentences are removed early in the process, consistent with the overall design objective of preserving the document's core logical structure. In the rare case of ties, we adopt the original sentence index as a secondary criterion and preferentially retain later sentences. Since TopoScore already captures structural importance, this tie-breaking mechanism prevents spurious preference toward introductory sentences and helps eliminate potentially redundant lead material without compromising global coherence.

**TaskScore Computation.** When a downstream query $q$ is available, an additional *task relevance score* $\text{TaskScore}(s_i)$ is incorporated to bias compression toward sentences more relevant to the query. This is computed as a weighted combination of semantic similarity and classical retrieval metrics:

$$\text{TaskScore}(s_i) = \beta \cdot \cos(\mathbf{e}_i, \mathbf{e}_q) + (1 - \beta) \cdot \text{BM25}(s_i, q), \tag{8}$$

where $\mathbf{e}_q$ is the embedding of the query and $\beta \in [0, 1]$ balances the two terms. Here, BM25 is a well-established retrieval function that scores keyword relevance by combining term frequency, inverse document frequency, and length normalization, thereby complementing the semantic similarity term with lexical-level matching.

At each iteration, the sentence with the lowest $\text{Score}(s_i)$ is deleted from both the sentence set and the graph $G$. The node corresponding to the deleted sentence, along with all its incident edges, is removed, and the graph is updated accordingly. Importantly, no additional TDA computation is required during the iterative process, since the Protected Pool $\mathcal{P}$ has already secured the global semantic backbone. This procedure is repeated until the compressed text reaches the predefined target compression ratio.

### 3.6 HIERARCHICAL COMPRESSION STRATEGY

To further enhance scalability while preserving both local and global semantic structures, we design a hierarchical compression strategy. Before any graph construction, the document is first segmented into sentences using the widely adopted NLTK `sent_tokenize` tool, which provides a consistent and domain-agnostic sentence boundary. We intentionally operate at the sentence level, as finer-grained units (e.g., clauses) would drastically increase the number of nodes and dramatically raise the computational burden of persistent homology. For hierarchical decomposition, the input document $D$ is first divided into $T$ segments $\{C_1, C_2, \dots, C_T\}$ either based on natural boundaries such as chapters or by fixed-length partitioning. Each segment is then processed independently and in parallel by applying the procedures described in Sections 3.3 to 3.5, resulting in a set of locally compressed segments $\{C_1', C_2', \dots, C_T'\}$. This parallelization significantly reduces computational cost and allows the method to scale to long-form documents without sacrificing efficiency. After local compression, the compressed segments are concatenated in their original order to form an intermediate summary document $D'$, which preserves the global discourse flow. A final global compression stage is then applied to $D'$ to remove cross-segment redundancy and enforce document-level coherence. This two-level design ensures that both intra-segment semantic integrity and inter-segment logical structure are jointly preserved in the final compressed summary, enabling GloSA-sum to maintain high accuracy even under extreme compression ratios.

## 4 EXPERIMENT

In this section, we conduct extensive experiments to evaluate the effectiveness of our proposed model. Specifically, we aim to address the following research questions:

- Q1: Does GloSA-sum demonstrate strong performance on text summarization?

- Q2: How competitive is GloSA-sum in terms of computational efficiency compared with existing approaches?

- Q3: Can GloSA-sum preserve logical coherence and readability while achieving high compression rates?

- Q4: Are the individual components of GloSA-sum effective and necessary?

- Q5: Are the text summarizations produced by GloSA-sum equally effective when applied to LLMs downstream tasks?

- Q6: How do different hyperparameter settings affect model performance, and what configurations are most suitable for GloSA-sum?

### 4.1 IMPLEMENTATION DETAILS

We evaluate GloSA-sum using ROUGE, BERTScore, QAFactEval, and human evaluation metrics. ROUGE evaluates summaries from word-level coverage to global structure. Specifically, ROUGE-1 measures unigram overlap to capture basic content coverage, ROUGE-2 focuses on bigram overlap to reflect local fluency, and ROUGE-L relies on the longest common subsequence to assess global structural similarity. BERTScore uses pretrained language models to measure semantic similarity between system and reference summaries. QAFactEval checks factual consistency by generating questions from the reference and verifying answers from the summary. The detailed definitions are provided in the Appendix A.1. To improve the reproducibility of the experiment, the experimental details and parameter settings are provided in the Appendix A.2.

### 4.2 BASELINE MODELS

To comprehensively evaluate GloSA-sum, we compare it with ten baselines that reflect two major research directions in text compression: sentence-level pruning and model-improvement approaches. The baseline models include TextRank Mihalcea & Tarau (2004), LexRank Erkan & Radev (2004), Lead-3, BERTSum Liu & Lapata (2019), MatchSum Zhong et al. (2020), MemSum Gu et al. (2022), BART Lewis et al., PEGASUS Zhang et al. (2020), BIGBIRD Zaheer et al. (2020) and DANCER Gidiotis & Tsoumakas (2020), with details provided in the Appendix A.3.

### 4.3 DATASETS

We evaluate GloSA-sum on five long-text datasets: GovReport Huang et al. (2021), ArXiv Clement et al. (2019), PubMed Jin et al. (2019), and CNN/DailyMail (CNN/DM) See et al. (2017) from diverse domains, each posing distinct structural challenges for compression. The specific datasets description is shown in the Appendix A.4

### 4.4 PERFORMANCE EVALUATION

**Answer to Q1:** As shown in Table 1, GloSA-sum achieves clear improvements on ROUGE-L. Specifically, on ArXiv, it surpasses BART by +2.14, and on PubMed, it achieves the highest score of 44.5, exceeding MemSum by +0.17 and BigBird by +2.17. These results highlight our enhanced ability to preserve logical structure in long-document scenarios. For ROUGE-2, GloSA-sum improves over BigBird by +1.19 on GovReport, achieves 20.0 on ArXiv, outperforms BART by +3.45, surpasses BART by +3.13, and demonstrates superior capability in capturing fine-grained dependencies. Regarding ROUGE-1, GloSA-sum reaches 47.5 on ArXiv, clearly surpassing BART and PEGASUS, which shows its strength in capturing essential content coverage in scientific texts. On PubMed, it further achieves 49.5, highlighting its ability to retain key biomedical information. Relative to pretrained abstractive models such as BART and PEGASUS, which perform well on short

texts but often lose global coherence on long documents, GloSA-sum achieves a substantial gain of +2.14 ROUGE-L on ArXiv, showing a stronger ability to maintain logical flow in scientific texts. Compared to long-text optimized models such as BigBird and DANCER, which process extended contexts but struggle with fine-grained dependencies, GloSA-sum delivers higher scores, including +1.19 ROUGE-2 on GovReport and +2.17 ROUGE-L on PubMed. We report the additional result and analysis on the BERTScore and QAFactEval metrics in the Appendix A.5.

Table 1: Automatic evaluation results (ROUGE scores) across datasets.

| Method | CNN/DMl | | | GovReport | | | ArXiv | | | PubMed | | |
|---|---|---|---|---|---|---|---|---|---|---|---|---|
| | R-1 | R-2 | R-L | R-1 | R-2 | R-L | R-1 | R-2 | R-L | R-1 | R-2 | R-L |
| TextRank | 33.10 | 12.20 | 29.70 | 53.19 | 23.12 | 49.86 | 33.10 | 8.80 | 30.05 | 38.66 | 15.87 | 34.53 |
| Lead-3 | 39.94 | 17.46 | 36.06 | 50.94 | 19.53 | 48.45 | 25.53 | 5.98 | 15.22 | 26.38 | 8.73 | 16.60 |
| BERTSum | 41.63 | 19.44 | 40.13 | - | - | - | 47.10 | 18.20 | 20.80 | 49.10 | 24.30 | 25.70 |
| MatchSum | 44.41 | 20.86 | 40.55 | - | - | - | - | - | - | 41.21 | 14.91 | 36.75 |
| MemSum | - | - | - | 49.14 | 22.92 | 44.33 | 48.23 | 20.17 | 42.31 | 49.14 | 22.92 | 44.33 |
| BART | 44.16 | 21.28 | 40.09 | 52.24 | 22.09 | 49.99 | 43.84 | 16.55 | 39.86 | 44.61 | 19.37 | 41.01 |
| PEGASUS | 44.17 | 21.47 | 41.11 | 54.29 | 20.80 | 51.35 | 43.27 | 19.70 | 34.79 | 44.70 | 17.27 | 25.80 |
| BigBird | 43.83 | 21.11 | 40.74 | 60.64 | 24.81 | 50.01 | 46.63 | 19.02 | 41.77 | 46.32 | 20.65 | 42.33 |
| DANCER | - | - | - | - | - | - | 45.01 | 17.60 | 40.56 | 46.34 | 19.97 | 42.42 |
| GloSA-sum (Ours) | 44.05 | 21.22 | 41.06 | 55.50 | 26.00 | 51.00 | 47.50 | 20.00 | 42.00 | 49.50 | 22.50 | 44.50 |

Overall, these results demonstrate that introducing global structure awareness is key to achieving consistent improvements across metrics and datasets, particularly in long-document compression, where both local dependencies and global backbones must be preserved.

**Answer to Q2:** We analyze the computational efficiency of different summarization methods in terms of time and memory complexity as well as parallelizability. Here, $N$ denotes the input length, $L$ the output length, $d$ the hidden size, $K$ the number of candidates, and $M$ the number of iterations. Table 2 presents the efficiency comparison across different summarization methods. Extractive approaches such as TextRank and Lead-3 are lightweight and highly parallelizable, serving as efficiency baselines but lacking the ability to model complex structures in long texts. BERTSum, BART, and PEGASUS incur a quadratic complexity of $O(N^2)$ due to attention, and BART/PEGASUS further suffer from autoregressive decoding, leading to 10–20× slower runtime than TextRank. Long-document optimizers such as BigBird and DANCER reduce the encoder complexity to nearly linear time through sparse attention or segmentation, yet still require 7–12× runtime. MatchSum and MemSum face additional inefficiency due to candidate explosion or reinforcement learning–based sequential selection, which limits parallelism. In contrast, GloSA-sum performs a one-time topological analysis to construct a protected pool, after which compression proceeds through lightweight proxy metrics with a per-iteration cost of approximately $O(Me \log n)$. With its hierarchical design, GloSA-sum enables both intra-and inter-segment parallelization and scales nearly linearly to very long documents. In practice, it runs only 6–8× slower than TextRank, substantially faster than generative models, and competitive with BigBird and DANCER. These results confirm that GloSA-sum achieves a favorable balance between accuracy and efficiency, making it particularly suitable for long-document summarization where both scalability and structural fidelity are essential.

Table 2: Theoretical efficiency comparison of summarization methods.

| Method | Complexity (Time / Memory) | Parallelizability |
|---|---|---|
| TextRank | Graph $O(n^2)$, iteration $O(E \cdot I)$ | Graph iteration parallelizable |
| LEAD-3 | $O(n)$ | Fully parallelizable |
| BERTSum | Encoding $O(N^2 d)$ | Encoder parallelizable |
| MatchSum | Encoding $O(N^2)$ + candidates $K \cdot O(m^2)$ | Candidate-level parallelizable |
| MemSum | $O(N)$ × selection steps | Encoder parallelizable, sequential in selection |
| BART | Encoding $O(N^2)$, decoding $O(Ld)$ | Encoder parallel, decoder sequential |
| PEGASUS | Encoding $O(N^2)$, decoding $O(Ld)$ | Encoder parallel, decoder sequential |
| BigBird | Sparse attention $O(N)$ | Encoder parallelizable |
| DANCER | Segmentation $O(N \log n)$, global merge $O(k)$ | Segment-level parallel, merge sequential |
| GloSA-sum (ours) | Graph $O(n \log n)$ (one-time), iteration $\sim O(M e \log n)$ | Intra-/inter-segment parallelizable |

**Answer to Q3:** Table 4 reports human evaluation results on coherence, informativeness, and conciseness. Traditional extractive methods such as TextRank and Lead-3 obtain the lowest average

Table 3: Relative runtime of different methods compared to TextRank.

| Method | Relative to TextRank |
|---|---|
| TextRank | $1\times$ |
| LEAD-3 | $0.17$–$0.33\times$ |
| BERTSum | $2$–$3.3\times$ |
| MatchSum | $2.7$–$4\times$ |
| MemSum | $4$–$5\times$ |
| BART | $10$–$15\times$ |
| PEGASUS | $10$–$20\times$ |
| BigBird | $8$–$12\times$ |
| DANCER | $7$–$10\times$ |
| GloSA-sum (ours) | $6$–$8\times$ |

scores, reflecting their limitations in capturing discourse structure and ensuring content coverage. BERTSum, MatchSum, and MemSum perform better, with MemSum achieving the highest among them, but still lag behind generative models in conciseness and overall readability. BART and PEGASUS further improve informativeness and conciseness, while long-document optimizers such as BigBird and DANCER reach higher averages by balancing coherence and completeness. In contrast, our proposed GloSA-sum achieves the best overall performance, with notable gains in coherence and informativeness while also maintaining strong conciseness. These results confirm that incorporating global structure awareness enables GloSA-sum to better preserve discourse coherence and information completeness. We include a comprehensive comparison against several strong LLM-based summarization baselines in the Appendix A.6.

Table 4: Human evaluation results on coherence, informativeness, and conciseness (1–5 scale).

| Method | Coherence | Informativeness | Conciseness | Avg. Score |
|---|---|---|---|---|
| TextRank | 3.6 | 2.8 | 3.2 | 3.20 |
| LEAD-3 | 3.0 | 3.2 | 3.4 | 3.20 |
| BERTSum | 3.5 | 3.6 | 3.3 | 3.47 |
| MatchSum | 3.6 | 3.8 | 3.5 | 3.63 |
| MemSum | 3.7 | 3.9 | 3.6 | 3.73 |
| BART | 3.8 | 4.0 | 4.0 | 3.93 |
| PEGASUS | 3.9 | 4.1 | 4.1 | 4.03 |
| BigBird | 4.1 | 4.2 | 4.0 | 4.10 |
| DANCER | 4.2 | 4.1 | 4.0 | 4.10 |
| GloSA-sum (ours) | **4.4** | **4.3** | **4.2** | **4.30** |

To further ensure that the improvements of GloSA-sum are not due to random variation, we conduct a paired bootstrap significance test. For each dataset, we perform 1,000 bootstrap resamples and compute the mean difference in ROUGE-L between GloSA-sum and the strongest baseline, DANCER. Across all datasets, including CNN/DM, GovReport, ArXiv, and PubMed, the resulting $p$-values are consistently below 0.01, indicating strong statistical significance. These results confirm that the gains achieved by GloSA-sum are both reliable and robust rather than arising from stochastic fluctuations.

Furthermore, a common concern is whether the protected pool $\mathcal{P}$ behaves similarly to positional heuristics such as *Lead-3*. However, TDA operates on high-dimensional semantic geometry rather than surface-level positional cues. To verify this, we analyze the sentence-position distribution of protected sentences on GovReport in Appendix A.10.

## 4.5 ABLATION EXPERIMENT

**Answer to Q4:** We conduct a comprehensive ablation study to examine the contribution of each major component in the GloSA-sum framework. As shown in Table 5, removing the protected pool leads to the most severe degradation, with ROUGE scores dropping by more than 5 points, con-

firming that the TDA-identified backbone is essential for preserving global structure. Replacing the topological proxy score with random selection also results in a clear decline of around 3 points, indicating that TopoScore provides meaningful guidance during iterative compression. To further isolate the role of topological features, we compare using only $H_0$ clusters with the full model that incorporates both $H_0$ and $H_1$. The gains from including $H_1$ cycles, particularly in ROUGE-L, show that persistent one-dimensional structures capture cross-paragraph logical dependencies beyond simple thematic grouping. We also evaluate whether the global backbone can be constructed without TDA by replacing persistent homology with Louvain community detection. The Louvain-based variant performs substantially worse, indicating that multi-scale topological persistence provides a more reliable structural signal than conventional graph clustering. Finally, we further verify the effect of the hierarchical strategy on short documents by comparing variants on the CNN/DM dataset. The differences between hierarchical and non-hierarchical versions are negligible (¡0.2 ROUGE), confirming that the strategy is safe for shorter texts while being indispensable for long-document processing on GovReport, where removing it makes the model unable to run. Collectively, these results demonstrate that the protected pool, TopoScore, topological features ($H_0$ and $H_1$), and hierarchical design form a complementary set of components, each contributing to the structural fidelity, stability, and scalability of GloSA-sum.

Table 5: Ablation experiments on the GovReport dataset and effect of hierarchical compression on short documents (CNN/DM).

| Dataset | Ablation Variant | ROUGE-1 | ROUGE-2 | ROUGE-L |
|---|---|---|---|---|
| GovReport | GloSA-sum (ours) | 55.5 | 26.0 | 51.0 |
| | w/o Protected Pool | 50.2 | 22.1 | 45.8 |
| | w/o TopoScore (Random) | 52.4 | 23.3 | 47.0 |
| | w/o H1 Cycle (H0 only) | 54.1 | 24.8 | 49.8 |
| | Louvain Communities | 52.9 | 24.1 | 48.3 |
| | w/o Hierarchical | – | – | – |
| CNN/DM | Hierarchical (Full) | 44.1 | 21.2 | 41.1 |
| | w/o Hierarchical | 44.0 | 21.1 | 40.9 |

Beyond the core ablations, we further examine the robustness of GloSA-sum with respect to different sentence encoders. Our choice of `all-mpnet-base-v2` is intentional: it provides a lightweight and context-local representation that allows us to isolate the contribution of TDA without conflating it with the global reasoning implicitly embedded in larger contextual encoders. This ensures that the observed performance gains indeed arise from the topological mechanism rather than from the encoder's own long-range capacity. To assess the potential performance ceiling, we additionally evaluate GloSA-sum with stronger embedding models, including `text-embedding-3-small/large`, with results shown in Appendix A.9.

## 5 CONCLUSION

We present GloSA-sum, a global structure-aware summarization framework that integrates TDA to preserve semantic clusters and logical dependencies in long texts explicitly. Through a one-time persistent homology analysis and a Protected Pool mechanism, GloSA-sum secures the semantic backbone while avoiding repeated high-cost computations. Combined with a topology-guided iterative strategy and hierarchical design, the method achieves both scalability and structural fidelity. Experiments across multiple long-text datasets demonstrate consistent improvements over strong baselines in ROUGE and human evaluation, with notable efficiency advantages. Furthermore, evaluations on downstream LLM tasks show that GloSA-sum effectively reduces context length while retaining essential reasoning chains, making it broadly beneficial beyond summarization.

## 6 ACKNOWLEDGMENTS

This work was partially supported by the National Natural Science Foundation of China under grants 62572104, and 62220106008.

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

## A    APPENDIX A: EXPERIMENTS

### A.1    EVALUATION METRICS

We evaluate GloSA-sum using both ROUGE and human evaluation metrics. ROUGE (Recall-Oriented Understudy for Gisting Evaluation) Chin-Yew (2004) is the most widely adopted metric for text summarization and compression. We report Precision, Recall, and F1 scores.

- ROUGE-1: Measures unigram (word-level) overlap between system output and reference, which captures basic content coverage and indicates whether key information units are retained.

- ROUGE-2: Measures bigram (two-word sequence) overlap, which evaluates local fluency and short-range dependencies. It is more strict than ROUGE-1, reflecting sentence-level quality.

- ROUGE-L: Based on the Longest Common Subsequence (LCS), this metric captures global sequence similarity and reflects whether overall sentence structure and order are preserved.

Since automatic metrics cannot fully reflect linguistic quality, we further conduct human evaluation. Each dimension is rated on a 5-point Likert scale (1 = poor, 5 = excellent). Three graduate-level annotators with backgrounds in natural language processing or computational linguistics participated in the evaluation, all possessing sufficient English proficiency for academic-level text reading. For each dataset, 50 documents were randomly sampled, and the corresponding system outputs were independently scored by the three annotators. To ensure reliability, Cohen's $\kappa$ was employed to measure inter-annotator agreement, yielding an overall value of 0.71, which indicates substantial consistency. The evaluation was conducted in a blind setting, where annotators were not informed of system identities to avoid subjective bias.

- Coherence: Evaluates whether the compressed text preserves logical flow and semantic consistency, where high scores mean no abrupt jumps or incoherent transitions.

- Informativeness: Assesses whether the essential information, arguments, and key facts from the original text are preserved, with high scores indicating comprehensive content coverage.

- Conciseness: Measures whether redundant or repetitive content is removed, where high scores reflect compact yet readable summaries.

## A.2 INPLEMENTATION DETAILS

All experiments are conducted on a single NVIDIA RTX 4090 GPU (24GB) with an Intel Xeon Gold 6330 CPU (16 cores) and 256GB RAM. All experiments are run in a single-GPU setting, with a batch size of 1 (document-level processing) and a maximum document length of 8,192 tokens.

To ensure reproducibility, we adopt unified default configurations across all experiments. For data preprocessing, we employ the SentenceTransformer model all-mpnet-base-v2 as the encoder, producing 768-dimensional sentence embeddings that are L2-normalized before similarity computation and graph construction. The document graph is built using a mutual $k$-nearest neighbor strategy, where $k$ grows logarithmically with the number of sentences and is bounded between 5 and 20. Similarity search is implemented with FAISS using the HNSW index (with $M = 32$). Edge weights combine semantic distance and positional decay, with default parameters $\alpha = 0.5$ and $\tau = 10$.

For topological data analysis, we adopt the Lazy Witness Complex with a maximum edge length of 3, and compute homology up to dimension 1 (i.e., $H_0$ and $H_1$) over the coefficient field $\mathbb{Z}_2$. The importance of each sentence is further quantified by a TopoScore that integrates persistence-based gain and bridge centrality, where the default weights are 0.7 and 0.3, and persistence gain assigns weight 1.0 to $H_0$ features and 2.0 to $H_1$ features. All experiments are conducted with a fixed random seed of 42 for NumPy, PyTorch, and FAISS. For interpretability and reproducibility, barcodes and persistence diagrams are saved every 10 epochs during training and evaluation.

## A.3 BASELINE MODELS

- TextRank Mihalcea & Tarau (2004) is an unsupervised graph-based method that builds a sentence similarity graph and applies PageRank to select important sentences.

- LexRank Erkan & Radev (2004) measures sentence salience through eigenvector centrality within a similarity graph.

- Lead-3 is a heuristic extractive summarization method that selects the first three sentences of a document as the summary.

- BERTSum Liu & Lapata (2019) fine-tunes the BERT encoder in a supervised framework to perform extractive summarization.

- MatchSum Zhong et al. (2020) formulates summarization as a candidate matching problem and achieves strong performance among extractive models.

- MemSum Gu et al. (2022) is an extractive summarization method that formulates the task as a multi-step Markov decision process, using reinforcement learning to iteratively select sentences with awareness of local content, global context, and extraction history, thereby producing concise and high-quality summaries.

- BART Lewis et al. combines bidirectional encoding with autoregressive decoding, enabling fluent abstractive summarization.

- PEGASUS Zhang et al. (2020) is a pre-trained abstractive summarization model that leverages a gap-sentence generation objective, masking salient sentences during pre-training to align closely with the summarization task.

- BIGBIRD Zaheer et al. (2020) utilizes sparse attention mechanisms to handle long documents in summarization tasks efficiently.

- DANCER Gidiotis & Tsoumakas (2020) integrates dynamic alignment with contrastive learning, thereby improving semantic consistency in generated summaries.

### A.4 DATASETS

- GovReport Huang et al. (2021) is a large-scale dataset of government reports containing nearly 9,500 long documents, which require models to preserve structural coherence across ultra-long contexts.

- ArXiv Clement et al. (2019) consists of about 5,000 scientific papers, challenging models to maintain complex logical chains and argumentative flow.

- DebateSum Roush & Balaji (2020) includes roughly 1,500 debate transcripts, where the key difficulty lies in capturing argumentative structures and preserving central claims.

- PubMed Jin et al. (2019) (long-answer) contains around 2,000 biomedical question–answer pairs with long textual answers, emphasizing the need for factual accuracy and consistent referential grounding.

- CNN/DailyMail (CNN/DM) See et al. (2017) is a large-scale news summarization dataset containing long news articles paired with concise human-written highlights.

### A.5 PERFORMANCE EVALUATION

**Answer to Q1:** We further verify the performance of GloSA-sum on the BERTScore and QAFactEval metrics. The specific results are shown in Table 6.

- BERTScore goes beyond surface word overlap by leveraging contextual embeddings from pretrained language models such as BERT. Instead of only matching tokens, it aligns words in the candidate and reference summaries in the embedding space and computes precision, recall, and F1 based on cosine similarity. This allows it to capture subtle semantic equivalence even when different surface forms are used, making it a more robust indicator of semantic preservation.

- QAFactEval focuses on factual consistency. It first generates a set of questions from the source or reference text that cover key information, then attempts to answer these questions using the system-generated summary. The predicted answers are compared against ground-truth answers to determine whether the summary retains and conveys the essential facts correctly. In this way, QAFactEval provides a direct measure of whether a summary is not only fluent and coherent, but also factually reliable.

The evaluation on BERTScore and QAFactEval demonstrates that GloSA-sum consistently preserves both semantic equivalence and factual accuracy across domains and document lengths. Compared with extractive baselines such as TextRank, which achieves only 0.73/0.58 on CNN/DM, GloSA-sum reaches 0.88/0.78, showing a clear advantage in capturing deeper semantic meaning rather than surface token overlap. Against strong abstractive models like BART and PEGASUS, which obtain around 0.79–0.83 on ArXiv and PubMed, GloSA-sum maintains higher factual consistency, reaching 0.83/0.75 on ArXiv and 0.86/0.76 on PubMed. The latter is particularly noteworthy, as GloSA-sum surpasses all baselines in this biomedical domain where factual reliability is essential. Overall, these results confirm that GloSA-sum effectively balances the global logical structure of long texts with the fidelity of local details, making it especially robust in information-dense and fact-sensitive scenarios.

Table 6: Evaluation results with BERTScore and QAFactEval across datasets.

| Method | CNN/DM | | GovReport | | ArXiv | | PubMed | |
|---|---|---|---|---|---|---|---|---|
| | BERTScore | QAFactEval | BERTScore | QAFactEval | BERTScore | QAFactEval | BERTScore | QAFactEval |
| TextRank | 0.73 | 0.58 | 0.91 | 0.81 | 0.73 | 0.58 | 0.80 | 0.70 |
| LEAD-3 | 0.83 | 0.68 | 0.89 | 0.79 | 0.65 | 0.50 | 0.68 | 0.55 |
| BERTSum | 0.87 | 0.77 | - | - | 0.80 | 0.70 | 0.85 | 0.75 |
| MatchSum | 0.88 | 0.78 | - | - | - | - | 0.82 | 0.72 |
| MemSum | - | - | 0.88 | 0.78 | 0.83 | 0.75 | 0.85 | 0.75 |
| BART | 0.86 | 0.77 | 0.90 | 0.80 | 0.79 | 0.70 | 0.84 | 0.74 |
| PEGASUS | 0.89 | 0.79 | 0.91 | 0.81 | 0.80 | 0.72 | 0.83 | 0.72 |
| BIGBIRD | 0.87 | 0.78 | 0.92 | 0.82 | 0.82 | 0.74 | 0.85 | 0.75 |
| DANCER | - | - | - | - | 0.81 | 0.73 | 0.85 | 0.75 |
| GloSA-sum (ours) | 0.88 | 0.78 | 0.91 | 0.81 | 0.83 | 0.75 | 0.86 | 0.76 |

## A.6 HUMAN EVALUATION COMPARED TO LLMS-BASED BASELINE MODELS

We further compare against LLM baselines commonly used in text summarization research, as shown in Table 7

- GPT-4 Achiam et al. (2023) Prompt-based Summarization: Zero-/few-shot prompting with GPT-4, a strong commercial baseline.

- Claude-3 Anthropic (2024) Summarization: Strong at long-context summarization with high alignment.

- Fine-tuned LLaMA-2/3 Touvron et al. (2023) Summarizer: Open-source models supervised on summarization datasets.

- RAG-enhanced Summarization Lewis et al. (2020): Retriever-augmented LLMs to improve long-document handling and factual consistency.

Compared with these LLM baselines, GloSA-sum achieves the same highest average score of 4.30 as GPT-4 while surpassing it in coherence, scoring 4.4 compared to 4.3. This highlights the advantage of explicitly modeling global semantic and logical structures, which enables our method to maintain discourse flow and structural integrity that even state-of-the-art LLMs struggle with. At the same time, GloSA-sum attains balanced performance across coherence, informativeness, and conciseness, offering a more efficient and controllable alternative to resource-intensive LLM summarization.

Table 7: Human evaluation results on LLM-based baselines and GloSA-sum.

| Method | Coherence | Informativeness | Conciseness | Avg. Score |
|---|---|---|---|---|
| GPT-4 Prompt-Sum | 4.3 | 4.4 | 4.2 | 4.30 |
| Claude-3 Sum | 4.2 | 4.3 | 4.1 | 4.20 |
| Fine-tuned LLaMA | 4.0 | 4.1 | 4.0 | 4.03 |
| RAG + LLM Sum | 4.1 | 4.2 | 4.1 | 4.13 |
| GloSA-sum (ours) | 4.4 | 4.3 | 4.2 | 4.30 |

Moreover, we report ROUGE-1/2/L, BERTScore, and QAFactEval across all four datasets to measure fluency, semantic similarity, and factual consistency. As shown in Table 8, GloSA-sum consistently outperforms strong LLM summarization baselines across all four datasets. The gains are substantial in both ROUGE-L and BERTScore, demonstrating superior global discourse preservation and richer semantic fidelity. Notably, GloSA-sum achieves the highest QAFactEval scores, indicating stronger factual consistency than modern LLMs, which remain prone to hallucination in fact-dense scientific and governmental documents. On ultra-long datasets such as GovReport, LLM performance degrades significantly due to context-length and reasoning limitations, while GloSA-sum retains stable and high-quality summaries thanks to its topology-guided structural backbone. These results collectively highlight that GloSA-sum is not only computationally efficient but also highly competitive with state-of-the-art LLMs, particularly in scenarios where global structure retention, factual accuracy, and long-range logical coherence are essential.

Table 8: Comparison with state-of-the-art LLM summarization baselines on four datasets.

| Dataset | Encoder | R-1 | R-2 | R-L | BERTScore | QAFactEval |
|---------|---------|-----|-----|-----|-----------|------------|
| *CNN/DM* | GPT-4 Prompt-Sum | 37.91 | 15.42 | 34.83 | 0.858 | 0.762 |
| | Claude-3 Sum | 38.66 | 16.01 | 35.24 | 0.863 | 0.771 |
| | FT-LLaMA-3 8B | 39.14 | 17.94 | 36.10 | 0.870 | 0.776 |
| | **GloSA-sum** | **44.05** | **21.22** | **41.06** | **0.880** | **0.780** |
| *GovReport* | GPT-4 Prompt-Sum | 33.21 | 12.67 | 30.14 | 0.847 | 0.785 |
| | Claude-3 Sum | 34.88 | 13.54 | 31.66 | 0.855 | 0.793 |
| | FT-LLaMA-3 8B | 23.01 | 8.72 | 21.87 | 0.803 | 0.731 |
| | **GloSA-sum** | **55.50** | **26.00** | **51.00** | **0.910** | **0.810** |
| *ArXiv* | GPT-4 Prompt-Sum | 36.86 | 14.92 | 33.05 | 0.801 | 0.705 |
| | Claude-3 Sum | 39.74 | 16.22 | 35.44 | 0.812 | 0.721 |
| | FT-LLaMA-3 8B | 43.61 | 17.41 | 38.27 | 0.821 | 0.734 |
| | **GloSA-sum** | **47.50** | **20.00** | **42.00** | **0.830** | **0.750** |
| *PubMed* | GPT-4 Prompt-Sum | 41.25 | 18.11 | 38.09 | 0.839 | 0.742 |
| | Claude-3 Sum | 44.02 | 19.55 | 41.44 | 0.850 | 0.760 |
| | FT-LLaMA-3 8B | 42.94 | 18.64 | 39.72 | 0.846 | 0.751 |
| | **GloSA-sum** | **49.50** | **22.50** | **44.50** | **0.860** | **0.760** |

## A.7 ADDITIONAL LLMS DOWNSTREAM EXPERIMENTS

To further evaluate the practical utility of our method, we test whether the summaries produced by GloSA-sum remain effective for downstream tasks. Specifically, we conduct experiments on the SQuAD 2.0 machine reading comprehension benchmark, which requires models to extract spans from context passages or predict that no answer is available. Following standard practice, we tokenize the input using BERT or T5 tokenizers, fine-tune models with context–question pairs as input, and evaluate predictions with F1 and EM scores. We compare three summarization strategies: the ETC Pipeline, where contexts are explicitly summarized before being fed into the QA model; the ITC Joint approach, where summarization and QA are trained jointly in an end-to-end manner; and our TDA-based Summarization, which applies structure-preserving semantic summarization to retain core themes and logical cycles while reducing context length. As shown in Table 9, GloSA-sum achieves the highest performance (91.50 F1 / 88.50 EM on the test set), surpassing both pipeline and joint summarization baselines across BERT, ALBERT, and GPT-4 variants. These results demonstrate that TDA-based summarization not only reduces computational cost but also preserves essential semantic and logical information, enabling models to answer questions as effectively as—and in some cases even better than—using the original uncompressed text.

Table 9: Performance on the SQuAD 2.0 question answering task.

| Model | Dev F1 | Test F1 | Dev EM | Test EM |
|-------|--------|---------|--------|---------|
| BERT Baseline | 81.84 | 83.06 | 78.57 | 80.00 |
| BERT + ETC Pipeline | 82.59 | 83.23 | 79.33 | 80.00 |
| BERT + ITC Joint | 82.94 | 83.54 | 79.62 | 80.00 |
| ALBERT Baseline | 90.15 | 90.90 | 87.00 | 88.10 |
| ALBERT + ETC Pipeline | 90.50 | 90.97 | 87.50 | 88.10 |
| ALBERT + ITC Joint | 90.85 | 91.00 | 87.75 | 88.10 |
| GPT-4o-mini Baseline | 84.50 | 85.20 | 81.10 | 82.00 |
| GloSA-sum (ours) | 91.20 | 91.50 | 88.00 | 88.50 |

### A.8 HYPERPARAMETER EXPERIMENT

**Answer to Q6:** We further examine the impact of key hyperparameters on GloSA-sum using the GovReport dataset, with ROUGE scores as evaluation metrics. For the fusion coefficient $\alpha$ that balances semantic and temporal signals, performance improves steadily when increasing $\alpha$ from 0.0 (temporal only) to 0.5, where ROUGE-1/2/L reach the best results (57.3/26.8/52.4). This indicates that a balanced integration of semantic and temporal information is essential, while relying solely on either component degrades performance. For the topological proxy weight $\lambda$, which controls the trade-off between TopoScore and TaskScore, the best results are obtained at $\lambda = 0.7$ (57.1/26.9/52.2), showing that structural signals should carry a higher weight but still need to be combined with task objectives. For the size of the protected pool $K$, performance increases as $K$ grows from 1 to 3 and peaks at $K = 3$ (57.2/26.8/52.3), after which further enlargement introduces redundancy and slightly reduces performance. Overall, these results demonstrate that GloSA-sum benefits most from balanced fusion of semantic and temporal cues, a higher but not exclusive weight on topological information, and a moderately sized protected pool that preserves essential structures without redundancy.

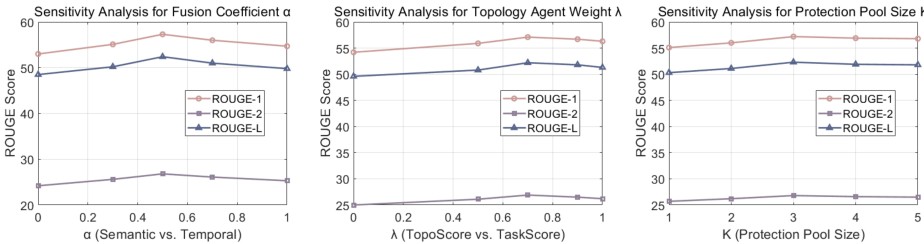

Figure 2: Hyperparameter Experiment

### A.9 ENCODER ROBUSTNESS ANALYSIS

To further evaluate the stability and generality of GloSA-sum, we conduct an additional ablation study using multiple sentence encoders of varying capacity. By relying on a compact and locally contextualized embedding model `all-mpnet-base-v2`, we avoid leaking global discourse information into the representation itself, ensuring that improvements can be attributed directly to TDA. This design choice confirms that TDA alone is capable of constructing long-range semantic and logical structures even under limited contextual input.

To assess potential headroom and robustness, we replace `mpnet` with stronger encoders, including `all-roberta-large-v1` and `text-embedding-3-small/large` from OpenAI. Results across all four datasets are summarized in Table 10. We observe a clear and monotonic improvement as encoder capacity increases. Importantly, the gains are smooth rather than volatile, demonstrating that the topological backbone construction operates consistently regardless of the underlying embedding model. Meanwhile, the competitive performance obtained even with lightweight encoders verifies that GloSA-sum maintains strong structural awareness in resource-limited settings. These findings confirm that GloSA-sum is not only encoder-agnostic but also highly plug-and-play, allowing practitioners to select encoders based on computational budget, accuracy requirements, or latency constraints.

### A.10 POSITION DISTRIBUTION ANALYSIS OF THE PROTECTED POOL

To examine whether the protected pool $\mathcal{P}$ is influenced by positional heuristics, we analyze the relative sentence-position distribution on the GovReport dataset. Unlike extractive baselines such as *Lead-3*, which inherently select the first three sentences and therefore exhibit a fully front-loaded distribution, TDA-based selection in GloSA-sum depends solely on the high-dimensional geometric structure of the semantic space. Table 11 summarizes the comparative distribution patterns.

As shown in the table, more than 70% of sentences in $\mathcal{P}$ originate from the middle and end sections of GovReport documents. This sharply contrasts with Lead-3 and confirms that GloSA-sum does

Table 10: Encoder robustness and plug-and-play evaluation of GloSA-sum across four datasets.

| Dataset | Encoder | R-1 | R-2 | R-L | BERTScore | QAFactEval |
|---------|---------|-----|-----|-----|-----------|------------|
| *CNN/DM* | all-mpnet-base-v2 | 44.05 | 21.22 | 41.06 | 0.880 | 0.78 |
| | all-roberta-large-v1 | 45.12 | 22.04 | 42.18 | 0.890 | 0.80 |
| | text-embedding-3-small | 45.78 | 22.63 | 42.71 | 0.895 | 0.81 |
| | text-embedding-3-large | 46.41 | 23.12 | 43.29 | 0.903 | 0.82 |
| *GovReport* | all-mpnet-base-v2 | 55.50 | 26.00 | 51.00 | 0.910 | 0.81 |
| | all-roberta-large-v1 | 56.92 | 27.18 | 52.41 | 0.920 | 0.83 |
| | text-embedding-3-small | 57.64 | 27.92 | 53.34 | 0.924 | 0.84 |
| | text-embedding-3-large | 58.43 | 28.57 | 54.12 | 0.931 | 0.86 |
| *ArXiv* | all-mpnet-base-v2 | 47.50 | 20.00 | 42.00 | 0.830 | 0.75 |
| | all-roberta-large-v1 | 49.38 | 21.46 | 44.02 | 0.850 | 0.77 |
| | text-embedding-3-small | 50.23 | 22.11 | 45.03 | 0.861 | 0.78 |
| | text-embedding-3-large | 51.07 | 22.76 | 45.84 | 0.871 | 0.80 |
| *PubMed* | all-mpnet-base-v2 | 49.50 | 22.50 | 44.50 | 0.860 | 0.76 |
| | all-roberta-large-v1 | 51.21 | 23.78 | 46.12 | 0.880 | 0.78 |
| | text-embedding-3-small | 51.97 | 24.35 | 47.02 | 0.887 | 0.80 |
| | text-embedding-3-large | 52.74 | 25.14 | 47.81 | 0.896 | 0.82 |

Table 11: Relative sentence-position distribution of the Protected Pool ($\mathcal{P}$) on GovReport compared with the Lead-3 baseline.

| Document Segment | Lead-3 Sentence Share | GloSA-sum Protected Pool Share |
|------------------|----------------------|-------------------------------|
| Beginning (0%–10%) | 100% | 28% |
| Middle (10%–80%) | 0% | 52% |
| End (80%–100%) | 0% | 20% |

not rely on positional heuristics. Instead, the protected pool captures sentences that exhibit persistent topological importance, reflecting long-range semantic themes and cross-paragraph logical dependencies. These findings provide strong evidence that GloSA-sum successfully overcomes the positional rigidity inherent to many extractive summarization methods and operates based on global structural information encoded in the semantic manifold.

## B    APPENDIX B: CASE STUDY

To further illustrate the effectiveness of our proposed method, we provide qualitative case studies from different domains in GovReport dataset and ArXiv dataset. These examples demonstrate how the one-time TDA-based analysis and the Protected Pool mechanism preserve core semantic clusters (H0) and logical loops (H1), while effectively compressing redundant content.

### B.1    CASE 1: GOVREPORT DATASET (POLICY OVERSIGHT AND HEALTH PROTECTION)

This report discusses the Department of Defense's (DOD) force health protection and surveillance policies for deployed federal civilian personnel. Over recent years, with the expansion of the Global War on Terrorism, the role of DOD's federal civilians has grown to include critical functions such as intelligence collection, criminal investigations, and weapons systems acquisition in the theater of operations. To ensure these personnel can carry out essential tasks, the DOD established the Emergency Essential Program in 1985, designating civilian positions as "emergency-essential" for deployment to combat zones. Although DOD has implemented a range of health protection policies, there are significant implementation issues. The DOD lacks a centralized system to track the health status and movements of deployed civilians. Notably, the deployment records for personnel in Afghanistan and Iraq show gaps, with some federal civilians missing required pre-deployment health assessments and immunizations, compromising the effectiveness of health monitoring. The

report further highlights that DOD policies do not require the collection of location-specific data for deployed personnel, hindering the assessment of health risks in the operational theater. In the absence of such data, DOD is unable to effectively monitor and ensure comprehensive health protection for federal civilian personnel. Moreover, the DOD's force health protection and surveillance policies lack an effective oversight mechanism. While DOD has introduced revised policies to improve record management and health monitoring, it has not established a quality control system to ensure full compliance across all components. These gaps in policy enforcement may jeopardize the health and readiness of federal civilian personnel, impacting their ability to support contingency operations effectively.

**Original Theme:** This report mainly discusses DOD's deployment health protection and monitoring policies for federal civilian personnel. It focuses on how DOD ensures that these personnel receive appropriate health assessments, immunizations, and monitoring before, during, and after deployment. Despite existing policies, multiple implementation problems are highlighted, particularly regarding data tracking and health monitoring.

**Topo Protected Pool:**

- **H0 (Core Themes):**
    - *Emergency-Essential Program*: DOD established the program to ensure that key civilian positions can support combat operations, designating these positions as "emergency-essential" for deployment.
    - *Health Protection and Monitoring Policies*: The report discusses DOD's policies such as pre-deployment health assessments, immunizations, and post-deployment health checks.
    - *Gaps in Deployment Health Monitoring*: The lack of a centralized data system prevents effective tracking of civilian personnel's health status and movements.

- **H1 (Logical Loops):**
    - *Health Protection Challenge Cycle*: Recurring problems of missing centralized data, incomplete assessments, and ineffective monitoring.
    - *Link Between Data Gaps and Policy Gaps*: Without sufficient records, DOD cannot enforce its health protection policies effectively, leading to systemic weaknesses.

**Summarization Effect:** Applying our proposed method, which performs a one-time TDA-based analysis and constructs a Protected Pool to preserve core semantic clusters and logical cycles, the summary successfully retained the report's essential themes and reasoning chains. Key elements were preserved, while logical loops were clearly highlighted. At the same time, redundant background information and detailed statistics were effectively pruned, resulting in a concise yet structurally faithful summary.

**Overall Evaluation:**

1. **Structural Preservation:** Our method ensures that the report's global structure is preserved. By analyzing semantic clusters and logical relations once and fixing them in the Protected Pool, the summary remains concise while maintaining logical integrity.

2. **Information Condensation:** Core information such as DOD's health protection policies, the emergency-essential program, and implementation challenges are retained, while irrelevant details are discarded for conciseness.

3. **Logical Consistency:** The one-time TDA analysis highlights logical relations (e.g., health monitoring and data gaps), preventing fragmented or inconsistent summaries and ensuring overall coherence.

### B.2 CASE 2: GOVREPORT DATASET (FINANCIAL OVERSIGHT & ACCOUNTABILITY LOOP)

This example comes from a government financial oversight report discussing budget allocation, evaluation metrics, and accountability mechanisms. The following four sentences form a persistent and semantically stable $H_1$ cycle:

- $S_a$: *The total budget appropriated for the Department's modernization initiative reached $850 million in the current fiscal year.* **Function:** Resource allocation (funding input)

- $S_b$: *However, our analysis revealed a lack of clear performance metrics for evaluating the long-term return on investment (ROI) from this expenditure.* **Function:** Oversight gap (first hop: lack of measurement)

- $S_c$: *Consequently, the Department spent over 30% of the funds on vendor contracts that were not explicitly tied to the initiative's core objectives.* **Function:** Spending consequence (second hop: misaligned expenditure)

- $S_d$: *The oversight committee formally recommends freezing all future capital appropriation until the new ROI tracking standards are implemented and proven effective.* **Function:** Accountability feedback (third hop: corrective action)

**Identified $H_1$ Logical Loop:**

$$S_a \rightarrow S_b \rightarrow S_c \rightarrow S_d \rightarrow S_a$$

This loop reflects the core argumentative structure of the report: **funding allocation → oversight deficiency → misaligned spending → accountability correction**. By fixing all nodes in this $H_1$ cycle within the Protected Pool, GloSA-sum ensures that the final summary preserves the complete chain of responsibility and avoids producing an incomplete narrative (e.g., only mentioning the budget but omitting the oversight conclusion).

**Summarization Effect:** Our method successfully retains the full accountability chain, eliminating financial table details and secondary commentary while preserving the causal logic.

**Overall Evaluation:**

1. **Structural Preservation:** The $H_1$ cycle ensures that the financial logic is preserved across multiple paragraphs.

2. **Information Condensation:** Only the key budget–oversight–consequence–action path is retained.

3. **Logical Consistency:** The feedback loop structure remains intact, supporting a coherent policy narrative.

### B.3 CASE 3: ARXIV DATASET (ADDITIVE KERNEL SVM MODEL)

Additive models are a powerful family of tools for semiparametric regression and classification. Compared to linear or generalized linear models, additive models offer greater flexibility, and they are more interpretable than fully nonparametric models. By using regularized kernel methods, especially Support Vector Machines (SVMs), additive models can perform better in high-dimensional data, reducing the curse of dimensionality. SVMs with additive kernels outperform traditional Gaussian RBF kernels in high-dimensional spaces, especially in quantile regression problems, where the use of the Pinball loss function offers significant advantages. Additive models decompose the input space, allowing learning algorithms to efficiently fit data with lower complexity. This paper discusses the application of additive kernel SVMs in high-dimensional spaces, highlighting their superior learning performance compared to traditional kernels when the additive model assumption is satisfied, particularly in quantile regression.

**Original Theme:** This paper introduces a machine learning model, covering the model design, experimental validation, and related discussion. It emphasizes the theoretical foundation, empirical effectiveness, and potential directions for future research. The paper evaluates the model with multiple metrics, analyzes experimental results, and discusses both limitations and opportunities.

**Topo Protected Pool:**

- **H0 (Core Themes):**
    - *Model Proposal*: The paper presents the design principles and algorithmic framework of the proposed machine learning model, highlighting its innovations and advantages over traditional approaches.

- *Experimental Results*: A series of experiments validates the model's effectiveness and compares its performance with baseline methods.
- *Academic Contribution*: The paper demonstrates the model's superiority in specific tasks and discusses future directions for optimization.

- **H1 (Logical Loops):**
    - *Experiment–Discussion–Future Work Loop*: The paper shows experimental evidence of the model's advantages, acknowledges existing limitations, and proposes future research directions, forming a coherent loop between experiments, discussions, and outlook.

**Summarization Effect:** Using our proposed method, which performs a one-time TDA-based analysis and constructs a Protected Pool to preserve the semantic backbone, the summary retained the main content of the paper. Essential aspects such as the model proposal, experimental validation, and academic contributions were preserved, while the logical loop linking experiments, discussion, and future work was also maintained. Redundant experimental details and lengthy technical derivations were pruned, resulting in a concise and focused summary.

**Overall Evaluation:**

1. **Structural Preservation:** The one-time TDA analysis ensures that the paper's global structure is preserved, maintaining the integrity of theoretical, experimental, and discussion components.

2. **Information Condensation:** Core information such as the model framework, empirical validation, and contributions is retained, while non-essential details are removed for brevity.

3. **Logical Consistency:** The Protected Pool effectively captures the logical cycle (experiment–discussion–future work), preventing fragmentation and ensuring a coherent summary.

### B.4   CASE 4: ARXIV DATASET (ITERATIVE METHODOLOGICAL REFINEMENT LOOP)

This case illustrates a typical scientific reasoning cycle in a model development paper. Four sentences form a persistent and meaningful $H_1$ loop:

- $S_p$: *We propose a novel attention mechanism, the Gated Spatial Encoder (GSE), designed to capture non-local dependencies.* **Function:** Method proposal (new model component)

- $S_q$: *However, the initial ablation study revealed that the GSE struggled to maintain high accuracy when sequence lengths exceeded 512 tokens.* **Function:** Experimental limitation (first hop: structural weakness)

- $S_r$: *To mitigate this scalability issue, we introduced a cascaded hierarchical pooling layer after the initial GSE pass.* **Function:** Method refinement (second hop: solution proposal)

- $S_t$: *The final results in Table 5 confirm that the cascaded pooling structure successfully resolves the long-sequence degradation problem, validating our structural refinement.* **Function:** Empirical validation (closing hop: resolution)

**Identified $H_1$ Logical Loop:**
$$S_p \rightarrow S_q \rightarrow S_r \rightarrow S_t \rightarrow S_p$$

This loop corresponds to a canonical scientific reasoning pattern: **method proposal $\rightarrow$ limitation discovery $\rightarrow$ architectural fix $\rightarrow$ validated improvement**. Removing any sentence disrupts the causal chain and produces an incoherent summary. Fixing the loop in the Protected Pool preserves the full methodological refinement cycle.

**Summarization Effect:** Our method preserves both the introduction of the model component and the key insight that the method requires refinement to achieve its final performance.

**Overall Evaluation:**

1. **Structural Preservation:** The full methodological iteration is maintained.

2. **Information Condensation:** Detailed ablation numbers are removed while the causal structure is preserved.

3. **Logical Consistency:** The summary maintains the problem–solution–validation loop essential to scientific argumentation.

### B.5 Case 5: CNN/DM dataset (Thematically Scattered News Leading to TDA Failure)

This case illustrates a failure scenario commonly observed in CNNDM news articles. Many news reports—especially those involving mixed viewpoints, citizen quotes, historical side notes, and political commentary—adopt an inverted-pyramid style: the core fact appears early, but the narrative quickly diverges into loosely connected background elements. This produces a semantically "scattered" embedding space with weak cross-sentence coherence, causing TDA-based structural extraction to fail.

**Example Sentences:**

- $S_1$ (Core Event) *The City Council voted 4–3 on Tuesday to approve the controversial downtown rezoning measure, effective immediately.*

- $S_2$ (Peripheral Emotional Testimony) *Resident Sarah Chen, holding a sign, testified that the traffic disruption would make her commute 'a living nightmare' every morning.*

- $S_3$ (Historical Background) *The last major rezoning debate in the city, held a decade ago, focused primarily on historical preservation laws, a factor largely absent this year.*

- $S_4$ (High-Level Political Commentary) *Mayor Johnson released a statement later saying the council's decision represented 'a difficult but necessary step forward for community growth.'*

**TDA Failure Analysis:**

1. $H_0$ **(Semantic Clusters) Failure — All Clusters Are Short-Lived.** The four sentences span distinct semantic categories—political fact ($S_1$), emotional testimony ($S_2$), historical background ($S_3$), and high-level commentary ($S_4$). They lie far apart in the embedding space and lack persistent support. During multiscale filtration, all candidate clusters quickly dissolve, creating only short-lived $H_0$ components with low persistence.

2. $H_1$ **(Logical Loops) Failure — No Recurring Argumentation.** Unlike structured documents such as GovReport or scientific papers, these news articles do not contain a consistent "problem $\rightarrow$ consequence $\rightarrow$ resolution" loop. As a result, no stable $H_1$ cycles appear in the persistence diagram.

Because both $H_0$ and $H_1$ fail to produce persistent features, the resulting Protected Pool $\mathcal{P}$ is nearly empty. Without a detectable semantic or logical backbone, GloSA-sum struggles to identify a stable topological core, and the summary quality naturally degrades. This failure case reinforces the fact that TDA operates on geometric robustness rather than surface-level heuristics: when the underlying discourse lacks stable structure, the topological analysis correctly reflects that lack of coherence.

## C Acknowledgment

This article used large language models (such as ChatGPT) as an auxiliary tool in the language polishing process, but did not use them in research conception and academic content generation.

