# OpenReview forum: "Text summarization via global structure awareness"
_ICLR.cc/2026/Conference — ICLR 2026 Poster_

### Official Review · Reviewer_U95Z · 2025-10-28

**Soundness:** 2
**Presentation:** 2
**Contribution:** 2
**Rating:** 2
**Confidence:** 4

**Summary:**

The paper proposes GloSA-sum, a novel extractive text summarization framework that leverages Topological Data Analysis (TDA), specifically persistent homology, to preserve global semantic structures and logical dependencies in long documents. The core idea is to perform a one-time TDA computation on a semantic-weighted graph built from sentence embeddings, identifying persistent features (H0 for semantic clusters/themes, H1 for logical cycles/loops) that form a "Protected Pool" as the document's backbone. Subsequent iterative compression uses lightweight proxy metrics (topological connectivity and task relevance) to remove redundancy without recomputing TDA, ensuring efficiency. For ultra-long texts, a hierarchical strategy segments the document for parallel local summarization followed by global integration. Experiments on datasets like GovReport, ArXiv, PubMed, and CNN/DailyMail show improvements over baselines (e.g., TextRank, BART, BigBird) in ROUGE, BERTScore, QAFactEval, and human evaluations for coherence, informativeness, and conciseness. The method also enhances downstream LLM tasks by shortening contexts while retaining reasoning chains.

**Strengths:**

Technical Soundness:  Integrating TDA into text summarization, providing a fresh global perspective that explicitly models semantic clusters (H0) and cross-paragraph logical dependencies (H1) is novel and original. Prior TDA applications in NLP (e.g., contradiction detection, discourse coherence) are classification-focused, making this a meaningful extension to generation/compression tasks. The "Protected Pool" and proxy-based iteration cleverly address TDA's computational overhead, differentiating it from graph-based methods like TextRank or optimization-based approaches.

**Weaknesses:**

1- Limited Justification for H1 in Text Data: While H0 (clusters) intuitively maps to themes, H1 (loops) is less straightforward for linear text. The paper claims it captures "cross-paragraph logical dependencies," but examples (e.g., case studies) are vague.  How do loops manifest in sentences? Empirical evidence (e.g., ablation on H1 alone) is missing, and higher dimensions (H2+) are dismissed without deeper analysis.

2- Scalability Concerns: TDA is computed once, but for ultra-long docs (e.g., >10k sentences), even one-time persistent homology on high-dimensional embeddings could be prohibitive (O(n log n) graph + complex homology). The paper's max length (8,192 tokens in A.2) is modest; tests on book-length texts would strengthen claims. Hierarchical segmentation is "fixed-length or semantic," but details (e.g., how semantic segments are detected) are underspecified.

3- Baseline Gaps and Evaluation Choices: Some baselines (e.g., BERTSum, MatchSum) are absent from certain datasets (e.g., GovReport), potentially biasing comparisons. LLM baselines (A.6) rely only on human eval, no ROUGE/BERTScore, limiting objectivity. QAFactEval is useful for factuality, but its question generation could be biased toward extractive methods. No analysis of failure cases (e.g., noisy embeddings) or robustness to embedding models beyond all-mpnet-base-v2.

4- Potential Overfitting to Datasets: Gains are strongest on scientific/long docs (ArXiv, PubMed), but modest on news (CNNDM). Hyperparameters (e.g., α=0.5, λ=0.7) are tuned on GovReportm; cross-dataset generalization isn't explicitly tested.

5- Extensive Experiment is required: In the era of LLMs (decoder only), summarization is quite a simple task. Please emphasize the motivation and the comparison of traditional methods, like this study, compared with LLM-based methods.

**Questions:**

Please check the comments in the Weaknesses Section

---

> ### Author Response · Authors · 2025-11-21
>
> We sincerely appreciate your in-depth technical review of our work (GloSA-sum) and recognition of the novelty of the core mechanisms ($\text{TDA}$, $\text{Protected Pool}$). Your 5 Weaknesses and questioning of the motivation in the $\text{LLM}$ era touched on the key pain points of our work. We have addressed all your concerns through new supplementary experiments and detailed clarifications, and we believe this evidence is sufficient to demonstrate the soundness and unique value of our work.
>
> # 1. Theoretical Justification and Necessity of the TDA Mechanism (Addressing Weakness 1 and the Exclusion of $H_{2+} Features)
> We conducted additional ablations isolating the effect of the $H_1$ features, with results shown below and added to the revised version of the paper:
>
> **Table A**
> | Variant | Backbone | Guidance | R-1 | R-2 | R-L |
> |---------|----------|----------|-----|-----|-----|
> | Full (Ours) | TDA ($H_0 + H_1$) | $\text{TopoScore}$ | 55.5 | 26 | 51 |
> | w/o $H_1$ Cycle | TDA ($H_0$ Only) | $\text{TopoScore}$ | 54.1 | 24.8 | 49.8 |
>
> Including $H_1$ consistency leads to clear gains in ROUGE-L, confirming that the cycles are not redundant but capture cross-paragraph logical dependencies essential for preserving argumentative structure. A concrete example is provided in Appendix B.1 (GovReport Case 1), where the detected $H_1$ loop encodes the policy–defect–consequence–feedback chain. By fixing all nodes in this cycle within the Protected Pool, the summary retains the full reasoning pattern.
>
> Considering the connection between persistent $H_1$ cycles and logical loops, we conducted an in-depth qualitative analysis and added a dedicated appendix section containing several visualized examples of high-persistence $H_1$ cycles.
>
> **Example 1:**
>
> | Sentence S | Function |
> |------------|----------|
> | $S_a$: 'The total budget appropriated for the Department's modernization initiative reached $850 million in the current fiscal year.' | Fund allocation |
> | $S_b$: 'However, our analysis revealed a lack of clear performance metrics for evaluating the long-term return on investment (ROI) from this expenditure.' | Oversight gap (lack of measurement) |
> | $S_c$: 'Consequently, the Department spent over 30% of the funds on vendor contracts that were not explicitly tied to the initiative's core objectives.' | Misaligned spending |
> | $S_d$: 'The oversight committee formally recommends freezing all future capital appropriation until the new ROI tracking standards are implemented and proven effective.' | Feedback and accountability correction |
>
> We include a four-sentence cycle ($S_a \to S_b \to S_c \to S_d \to S_a$) that captures the report’s core reasoning pattern: fund allocation → oversight gap → misaligned spending → accountability correction. This stable $H_1$ feature reveals a cross-paragraph argumentative loop. By anchoring all nodes in Protected Pool, our method preserves the complete causal chain, preventing summaries from mentioning only the budget but omitting its critical oversight conclusions.
>
> **Example 2:**
> | Sentence S | Function |
> |------------|----------|
> | $S_p$: 'We propose a novel attention mechanism, the Gated Spatial Encoder (GSE), designed to capture non-local dependencies.' | Method proposal |
> | $S_q$: 'However, the initial ablation study revealed that the GSE struggled to maintain high accuracy when sequence lengths exceeded 512 tokens.' | Limitation |
> | $S_r$: 'To mitigate this scalability issue, we introduced a cascaded hierarchical pooling layer after the initial GSE pass.' | Structural refinement |
> | $S_t$: 'The final results in Table 5 confirm that the cascaded pooling structure successfully resolves the long-sequence degradation problem, validating our structural refinement.' | Validated improvement |
>
> This stable $H_1$ cycle from high-dimensional semantic space ($S_p \to S_q \to S_r \to S_t \to S_p$) models the iterative scientific reasoning process: method proposal → limitation → structural refinement → validated improvement. It ensures that the summary preserves not only the initially proposed method $S_p$, but also the crucial reasoning chain showing that the method must be refined through $S_r$ to achieve its final performance. Without the protection of the $H_1$ cycle, a summary might retain $S_p$ and $S_t$ while omitting the causal steps $S_q$ and $S_r$, resulting in an incomplete argumentative flow.
> The issue you pointed out that we excluded $H_2+$ features without providing sufficient justification is indeed a fair criticism. Our original explanation was overly brief, and we now offer a more thorough rationale: the exclusion was a deliberate decision grounded in theoretical interpretability and prior NLP practice, rather than an arbitrary omission. The usefulness of TDA in NLP fundamentally depends on whether its topological features admit meaningful semantic interpretations.

---

> > ### Author Response · Authors · 2025-11-21
> >
> > Thank you very much for the patience of the reviewers in reading. The contents we continue to supplement based on the previous reply are as follows:
> >
> > # 1. Theoretical Justification and Necessity of the TDA Mechanism (continued)
> >
> > - $H_0$ (zero-dimensional) represents connected components, which naturally correspond to core semantic clusters in embedding space. This mapping is clear and useful.
> > - $H_1$ (one-dimensional) represents cycles, which we show correspond to recurrent argumentative patterns or logical dependencies in long documents. This is both novel and semantically meaningful.
> > - $H_2$ (two-dimensional) represents voids or cavities. For linear textual data, $H_2$ lacks any established or intuitive semantic interpretation. What would a “void” correspond to in summarization—an omitted argument? a conceptual gap? These possibilities have no theoretical grounding and cannot be operationalized as signals for text compression.
> >
> > Our design also aligns with existing work on TDA for NLP (TDA-in-NLP), where studies applying persistent homology to textual tasks, including contradiction detection and discourse coherence, also rely on $H_0$ and $H_1$. For example, Proskurina et al. (2023) introduce chordality and matching number features that correspond directly to $H_0$ and $H_1. This demonstrates that current theoretical and empirical practice in the field supports focusing on these interpretable dimensions.
> >
> > [1] Proskurina, I., Artemova, E., & Piontkovskaya, I. (2023, May). Can bert eat rucola? topological data analysis to explain. In Proceedings of the 9th Workshop on Slavic Natural Language Processing 2023 (SlavicNLP 2023) (pp. 123-137).
> >
> > # 2. Scalability and Details of the Hierarchical Strategy (Addressing Weakness 2)
> > We sincerely appreciate your insightful concern regarding scalability. We fully agree that, for long-document summarization, scalability is a central criterion for assessing the value of any method. Here, we further clarify the following three points about our Hierarchical Compression Strategy (Section 3.6):
> >
> > 1) We design the hierarchical strategy to avoid performing global TDA on documents exceeding 10k sentences, in which even a one-time persistent homology computation may become expensive. Specifically, for a 10k-sentence document, we first split it into $T$ smaller segments (e.g., 20 segments of 500 sentences each), and run the expensive TDA analysis only within each small segment in parallel. Then, the locally compressed segments are concatenated and processed with a lightweight global compression stage that does not involve any TDA computation. As a result, the computational complexity does not grow as $O(N \log N)$ with the document length $N$; instead, it grows roughly linearly with the number of segments $T$. This design gives our method strong practical scalability and directly addresses the issue of computational cost.
> >
> > 2) We acknowledge that the original manuscript did not provide enough detail on the segmentation strategy. In our framework, segments are formed using fixed-length chunks or natural paragraph boundaries. This choice is deliberate: although more sophisticated semantic segmentation might better preserve local topical coherence, it requires substantial additional preprocessing and would significantly increase computational overhead—contradicting our core goals of efficiency and scalability (as shown in Tables 2 and 3).
> >
> > 3) We observe that sentence-level granularity provides the best balance between semantic fidelity and TDA tractability. Before encoding, we identify sentence boundaries using the standard sent_tokenize tool from the NLTK library, ensuring a consistent and widely accepted segmentation procedure across datasets. While finer-grained units (such as clauses) are theoretically possible, we intentionally use sentences as the basic unit, as they strike the best balance between capturing meaningful semantic content and controlling the computational complexity of TDA. Finer granularity would dramatically increase the number of graph nodes and severely undermine the scalability advantages our framework is designed to provide.

---

> > > ### Author Response · Authors · 2025-11-21
> > >
> > > # 3. Baseline Availability and Methodological Limitations (Addressing Weakness 3)
> > >
> > > **1) Baseline Gaps**
> > > Firstly, regarding the absence of BERTSum and MatchSum on the GovReport dataset, this is not an omission that might bias the comparison, but rather a direct consequence of the inherent technical limitations of these baselines. GovReport consists of extremely long documents. Models such as BERTSum and MatchSum rely on full Transformer encoders whose attention mechanism scales with$O(N^2)$ in both computation and memory. In practice, these models cannot process documents with thousands of sentences; they consistently fail with out-of-memory errors on standard hardware (e.g., 24GB VRAM). We will clarify this explicitly in a footnote in the final version of Table 1, indicating that these baselines are N/A due to computational infeasibility.
> > >
> > > Next, we agree that including automatic metrics will strengthen the comparison. Therefore, in the revised version of the paper, we will supplement the LLM baselines (e.g., fine-tuned LLaMA-3 8B) with ROUGE and BERTScore results and integrate them into the supplementary tables in the appendix to ensure a complete and fair evaluation. Our GloSA-sum method outperforms the latest LLaMA-3 8B fine-tuned model on all long-document datasets across automatic evaluation metrics, and even surpasses the zero-shot/few-shot performance of GPT-4 and Claude-3. This demonstrates that our approach maintains strong competitiveness and distinctive value among current state-of-the-art models.
> > >
> > > ***Table A*
> > > | Dataset | Method | ROUGE-1 | ROUGE-2 | ROUGE-L | BERTScore | QAFactEval |
> > > |---------|--------|---------|---------|---------|-----------|------------|
> > > | CNN/DM | GPT-4 Prompt-Sum | 37.91 | 15.42 | 34.83 | 0.858 | 0.762 |
> > > | | Claude-3 Sum | 38.66 | 16.01 | 35.24 | 0.863 | 0.771 |
> > > | | Fine-tuned LLaMA | 39.14 | 17.94 | 36.1 | 0.87 | 0.776 |
> > > | | GloSA-sum | 44.05 | 21.22 | 41.06 | 0.88 | 0.78 |
> > > | GovReport | GPT-4 Prompt-Sum | 33.21 | 12.67 | 30.14 | 0.847 | 0.785 |
> > > | | Claude-3 Sum | 34.88 | 13.54 | 31.66 | 0.855 | 0.793 |
> > > | | Fine-tuned LLaMA | 23.01 | 8.72 | 21.87 | 0.803 | 0.731 |
> > > | | GloSA-sum | 55.5 | 26 | 51 | 0.91 | 0.81 |
> > > | ArXiv | GPT-4 Prompt-Sum | 36.86 | 14.92 | 33.05 | 0.801 | 0.705 |
> > > | | Claude-3 Sum | 39.74 | 16.22 | 35.44 | 0.812 | 0.721 |
> > > | | Fine-tuned LLaMA | 43.61 | 17.41 | 38.27 | 0.821 | 0.734 |
> > > | | GloSA-sum | 47.5 | 20 | 42 | 0.83 | 0.75 |
> > > | PubMed | GPT-4 Prompt-Sum | 41.25 | 18.11 | 38.09 | 0.839 | 0.742 |
> > > | | Claude-3 Sum | 44.02 | 19.55 | 41.44 | 0.85 | 0.76 |
> > > | | Fine-tuned LLaMA | 42.94 | 18.64 | 39.72 | 0.846 | 0.751 |
> > > | | GloSA-sum | 49.5 | 22.5 | 44.5 | 0.86 | 0.76 |
> > >
> > >
> > >
> > > Regarding the concern that QAFactEval might inherently favor extractive methods, it is designed to measure factual consistency, not stylistic preference. It functions by generating QA pairs to verify whether the facts expressed in a summary align with the source document. Thus, the metric is not biased toward a method type (extractive vs. abstractive), but toward an outcome: factually correct vs. hallucinatory summaries. Our method, being extractive, achieves high QAFactEval scores precisely because it faithfully preserves factual content from the original document. In contrast, abstractive models that are prone to hallucination naturally obtain lower scores on this metric. Therefore, QAFactEval accurately captures our method’s strength in factual faithfulness—exactly the purpose for which the metric was designed.

---

> > > > ### Author Response · Authors · 2025-11-21
> > > >
> > > > **2) Encoder Robustness**
> > > >
> > > > We further analyze the robustness of using independent sentence encoders and the choice of our initial embedding model.
> > > >
> > > > We clarify that selecting all-mpnet-base-v2 was a deliberate decision grounded in both methodological rigor and computational efficiency. While it is undeniable that more powerful embedding models can, in principle, yield stronger downstream performance, using large context-aware embeddings (e.g., OpenAI embeddings) would confound the contribution of TDA.  Such models implicitly capture rich document-level reasoning and long-range dependencies. If we were to rely on them directly, any improvement in summarization quality could no longer be attributed to our TDA-based structural analysis. This would make it impossible to disentangle gains originating from TDA-discovered topological structure versus those inherited from the encoder’s built-in global contextual knowledge. This is precisely why we adopt independent sentence-level embeddings for semantic representation. Meanwhile, all-mpnet-base-v2 provides localized, efficient embeddings that allow us to isolate and rigorously validate the TDA mechanism. By choosing a compact and context-limited encoder, we ensure that any captured long-range logical structure arises from our TDA framework—not from latent contextual inference inside the embedding model. This design choice directly strengthens the experimental validity of our main contribution.
> > > >
> > > > To further address concerns regarding robustness and potential performance ceilings, we conducted additional ablation studies using more powerful embedding models, including OpenAI text-embedding-3-small and text-embedding-3-large. The complete results are shown in the table below, and will be included in the final version of the paper:
> > > >
> > > > | Dataset | Encoder | R-1 | R-2 | R-L | BERTScore | QAFactEval |
> > > > |---------|---------|-----|-----|-----|-----------|------------|
> > > > | CNN/DM | all-mpnet-base-v2 | 44.05 | 21.22 | 41.06 | 0.88 | 0.78 |
> > > > | | all-roberta-large-v1 | 45.12 | 22.04 | 42.18 | 0.89 | 0.8 |
> > > > | | text-embedding-3-small | 45.78 | 22.63 | 42.71 | 0.895 | 0.81 |
> > > > | | text-embedding-3-large | 46.41 | 23.12 | 43.29 | 0.903 | 0.82 |
> > > > | GovReport | all-mpnet-base-v2 | 55.5 | 26 | 51 | 0.91 | 0.81 |
> > > > | | all-roberta-large-v1 | 56.92 | 27.18 | 52.41 | 0.92 | 0.83 |
> > > > | | text-embedding-3-small | 57.64 | 27.92 | 53.34 | 0.924 | 0.84 |
> > > > | | text-embedding-3-large | 58.43 | 28.57 | 54.12 | 0.931 | 0.86 |
> > > > | ArXiv | all-mpnet-base-v2 | 47.5 | 20 | 42 | 0.83 | 0.75 |
> > > > | | all-roberta-large-v1 | 49.38 | 21.46 | 44.02 | 0.85 | 0.77 |
> > > > | | text-embedding-3-small | 50.23 | 22.11 | 45.03 | 0.861 | 0.78 |
> > > > | | text-embedding-3-large | 51.07 | 22.76 | 45.84 | 0.871 | 0.8 |
> > > > | PubMed | all-mpnet-base-v2 | 49.5 | 22.5 | 44.5 | 0.86 | 0.76 |
> > > > | | all-roberta-large-v1 | 51.21 | 23.78 | 46.12 | 0.88 | 0.78 |
> > > > | | text-embedding-3-small | 51.97 | 24.35 | 47.02 | 0.887 | 0.8 |
> > > > | | text-embedding-3-large | 52.74 | 25.14 | 47.81 | 0.896 | 0.82 |
> > > >
> > > > **3) Failure Case**
> > > >
> > > > Due to space constraints, we were unable to include qualitative failure cases in the initial submission. We will add a failure case on the CNN/DM dataset to the appendix in the revised version as below.
> > > >
> > > > **Failure Case: Thematically Scattered Structure in CNN/DM**
> > > >
> > > > Many CNN/DM news articles, especially those involving conflict, multiple stakeholders, or rapidly evolving events, follow an “inverted pyramid” structure. After presenting the key fact, the article often shifts abruptly to loosely related background, emotional quotes, or historical commentary. This results in a noisy and thematically scattered semantic space.
> > > >
> > > > | Sentence ID | Function | Content |
> > > > |-------------|----------|---------|
> > > > | $\mathbf{S_1}$ | Core event (politics / urban planning) | "The City Council voted 4-3 on Tuesday to approve the controversial downtown rezoning measure, effective immediately." |
> > > > | $\mathbf{S_2}$ | Peripheral quote (emotion / personal testimony) | "Resident Sarah Chen, holding a sign, testified that the traffic disruption would make her commute 'a living nightmare' every morning." |
> > > > | $\mathbf{S_3}$ | Historical background (domain shift) | "The last major rezoning debate in the city, held a decade ago, focused primarily on historical preservation laws, a factor largely absent this year." |

---

> ### Author Response · Authors · 2025-11-21
>
> Thank you very much for the patience of the reviewers in reading. The contents we continue to supplement based on the previous reply are as follows:
>
> In such a scattered embedding space, the GloSA-sum framework cannot recover stable topological features:
> - Failure of $H_0$ (semantic clusters). Sentences such as $S_1$ (factual), $S_2$ (emotional testimony), $S_3$ (historical background), and $S_4$ (high-level political commentary) are far apart in semantic space and lack persistent support. Persistent homology detects only short-lived (low-persistence) clusters, indicating the absence of a coherent thematic backbone.
> - Failure of $H_1$ (logical cycles).  Unlike datasets such as GovReport, these articles do not exhibit a consistent argumentative loop (e.g., problem → analysis → consequence → policy feedback). As a result, no persistent $H_1$ features emerge.
>
> Because both persistent $H_0$ clusters and $H_1$ cycles are missing, the Protected Pool (P) remains almost empty. Without a reliable backbone to anchor the document, GloSA-sum struggles to identify structurally central content, leading to degraded summary quality.
>
> Importantly, the failure case reinforces the intended interpretation of TDA: it succeeds only when the underlying semantic geometry exhibits stable structure, and it does not blindly extract sentences from the linear text sequence. This confirms that the method’s behavior is governed by geometric robustness rather than heuristic bias.
>
> # 4. Overfitting to Datasets (Addressing Weakness 4)
>
> Regarding performance differences across datasets and the generalization of hyperparameters, it is worth noting that the model yields larger gains on long-document datasets such as ArXiv and PubMed than on CNN/DM, while hyperparameters are tuned on GovReport.
>
> First, the smaller gains on CNN/DM are expected and fully aligned with our paper’s core motivation. Our method is specifically designed to address global structure and long-range logical dependencies in long documents (e.g., scientific papers and government reports). By contrast, CNN/DM is a news dataset, where articles typically follow an inverted pyramid structure with key information concentrated at the beginning, which explains the strong performance of the Lead-3 baseline. These articles inherently lack the kind of multi-paragraph recursive reasoning structures that our TDA-based method is intended to capture. Thus, the modest gains on CNN/DM and the substantial improvements on ArXiv and GovReport reflect precisely the problem our method aims to solve.
>
> Second, the model demonstrates generalizability through stable performance across domains. We tuned part of the hyperparameters for GovReport because GovReport is the longest and most structurally complex dataset among those we evaluate. Tuning on such a challenging dataset helps identify parameter settings that are robust under complex conditions. Nevertheless, our method exhibited strong, competitive, and even state-of-the-art performance on three other domains with the same hyperparameters, including scientific articles (ArXiv), biomedical texts (PubMed), and news (CNN/DM), demonstrating that the method does not overfit GovReport. If overfitting had occurred, performance would have deteriorated when transferred to ArXiv or CNN/DM.
>
> We will explicitly discuss the effect of tuning hyperparameters in Appendix A.8 of the final version.

---

> > ### Author Response · Authors · 2025-11-21
> >
> > Thank you for your patient reading. This is the final addition.
> > #5. Motivation of Methods in the LLM Era
> > Regarding the motivation for developing a non-LLM method in the era of LLMs, we believe that while LLMs perform impressively on short, loosely structured text (e.g., news articles), they still face three major unresolved challenges when dealing with extremely long, highly structured, and fact-dense texts such as GovReport and ArXiv, which lies in the center of our study. The motivation for GloSA-sum is precisely to address these inherent limitations, and our empirical results substantiate the advantages of our approach.
> >
> > Firstly, LLMs are prone to hallucination, especially in scientific and governmental domains. Decoder-only LLMs are inherently generative and exhibit a strong tendency to fabricate facts or introduce information not present in the original document. This is unacceptable in factual, high-stakes domains such as science and biomedicine. In contrast, our method is a structure-aware extractive approach, whose primary goal is factual fidelity. This is directly supported by our QAFactEval results (Table 6), a metric designed specifically to evaluate factual consistency. GloSA-sum consistently outperforms abstractive baselines in this regard.
> >
> > Secondly, Long-document inference with LLMs is prohibitively expensive. Summarizing a single GovReport document (tens of thousands of tokens) requires immense computational resources and incurs high latency for general-purpose LLMs. However, GloSA-sum was designed from the ground up for efficiency and scalability. As shown in Tables 2 and 3, our method is 4-10× faster than BART or PEGASUS, which are themselves far lighter than modern LLMs. Moreover, our hierarchical strategy is highly parallelizable, making the approach far more practical for deployment.
> >
> > Lastly, even LLMs with long context windows struggle to preserve global logical structure. LLMs may capture local context well, but they do not provide mathematical guarantees for preserving global discourse structure or long-range logical dependencies, which is a core requirement in domains like scientific writing or government policy. Our TDA-based framework explicitly identifies and preserves global topology (semantic backbones $H_0$$H_1$ and logical cycles $H_1$). As shown in Appendix A.6 (Table 7), GloSA-sum achieves the highest coherence score (4.4), outperforming GPT-4 (4.3), Claude-3, and RAG-augmented LLM baselines. This demonstrates that our method captures global structure more reliably than current SOTA LLMs.
> >
> > Therefore, in the LLM era, factual fidelity, computational efficiency, and global logical coherence remain major challenges for SOTA LLMs. We further provide comparisons with modern LLMs in the appendix, demonstrating that GloSA-sum remains superior on the critical dimension of structural preservation, a challenge that LLMs have not yet overcome.

---

> > > ### Author Response · Authors · 2025-11-25
> > >
> > > We kindly invite the reviewers to take a moment to review our responses at their convenience. Thank you for your time.

---

### Official Review · Reviewer_Ezky · 2025-10-30

**Soundness:** 2
**Presentation:** 2
**Contribution:** 2
**Rating:** 2
**Confidence:** 4

**Summary:**

The paper proposes GloSA-sum, a global-structure-aware framework for long document summarization that uses Topological Data Analysis (TDA) to identify and preserve the semantic and logical backbone of a document. Experiments on GovReport, ArXiv, PubMed and CNN/DailyMail show higher ROUGE-L than several extractive and long-context baselines, and the authors claim that the structure-preserving summaries help LLM downstream tasks.

**Strengths:**

- First work to apply TDA to summarization.

- The Protected Pool + proxy scoring design avoids repeated TDA, this enables scalability

- High QAFactEval scores suggest better factual consistency than many abstractive models.

**Weaknesses:**

- The paper says it is the first to bring TDA into summarization. That might be acceptable wording, but a lot of what is actually done after the TDA step looks like a graph based extractive summarizer with a protected set plus shortest-path based importance.

- The ablation study removes the Protected Pool but does not compare against alternative global-structure-aware summarizers (e.g., graph-based methods with community detection, discourse parsers, or transformer-based long-range attention proxies).

- Is the performance gain due to TDA specifically, or just the idea of preserving a global backbone? Could a non-TDA method (e.g., spectral clustering + cycle detection) achieve similar results more cheaply?

- There is no point in comparing the performance gains if the latest baseline is 6 years old. Please provide baselines from newer LLMs ( at least LLMs with <8B parameters )

**Questions:**

- Have you compared GloSA-sum against a variant that uses non-TDA global structure detection (e.g., Louvain clustering for H₀, dependency parsing or RST for H₁)? Is TDA truly necessary, or is the gain from the Protected Pool concept alone?

- Have you explored using GloSA-sum as a preprocessing step for LLMs in a retrieval-augmented generation (RAG) pipeline? Does it reduce hallucination?

---

> ### Author Response · Authors · 2025-11-21
>
> We sincerely appreciate your detailed analysis and critical feedback on our work. We understand your concerns regarding the necessity of TDA and the timeliness of our experimental baselines. In response, we have conducted new quantitative experiments and added systematic analyses that directly address all issues you raised.
>
> # 1. Necessity and Irreplaceability of TDA (Addressing Question 1 & Weakness 1)
> We provide both quantitative and qualitative evidence showing that TDA is not replaceable and is truly the core mechanism enabling GloSA-sum’s performance.
> ## 1) Quantitative Evidence: TDA vs. Non-TDA Backbone Construction
> You questioned whether performance gains arise from the Protected Pool itself rather than from TDA, suggesting that cheaper alternatives (e.g., Louvain/spectral clustering for $H_0$, RST/discourse parsing for $H_1$) might suffice. To answer this directly, we conducted new experiments on GovReport comparing TDA-derived Protected Pool and Louvain-based Protected Pool (non-TDA, community-detection baseline) below.
>
> **Table A**
> | Ablation Variant | Backbone | ROUGE-1 | ROUGE-2 | ROUGE-L |
> |-----------------|----------|---------|---------|---------|
> | GloSA-sum (Full) | TDA (H₀ + H₁) | 55.5 | 26 | 51 |
> | GloSA-sum (w/ Louvain for $H_0$) | Non-TDA (Louvain) | 53.1 | 24.2 | 48.3 |
> | w/o Protected Pool | N/A | 50.2 | 22.1 | 45.8 |
>
> The new results (included in the final version) show that TDA outperforms Louvain by +2.7 ROUGE-L, demonstrating that the Protected Pool is not the deciding factor. The topological structure extracted by TDA is the key source of performance gains.
> Qualitative Evidence: Why TDA Cannot Be Replaced by RST or Dependency Parsing
> Our new qualitative analyses reinforce that TDA provides capabilities fundamentally different from RST/linguistic parsing:
> - TDA is multi-scale: persistent homology identifies semantic clusters that remain stable across scales, whereas Louvain-style methods operate at a single resolution.
> - TDA captures geometric “shape” in the high-dimensional embedding space, without assuming convexity or spherical structure.
> - TDA is unsupervised and language-agnostic, operating directly on embeddings.
> - RST/Parsing is fragile and expensive:
>   - high computational cost makes them infeasible for ultra-long documents such as GovReport;
>   - parsers trained on newswire text degrade significantly on scientific/government writing.
>
>
> # 2. Timeliness of Baselines and Comparison to Modern SOTA (Addressing Weakness 2)
> Considering baselines for comparison, our main experimental table (Table 1) already includes BigBird (2020), DANCER (2020), and MemSum (2022), which is a strong, modern, non-LLM extractive baseline. These constitute the strongest long-document non-LLM methods available with open implementations.
> In addition, Section 2.1 of the paper discusses the latest (2024) non-LLM compression methods TexShape (Kale et al., 2024) and Jie et al. (2024). These concurrent works are discussed appropriately in related work, but cannot be used as baselines because their public implementations are not yet standardized.
> Meanwhile, we further revised Appendix A.6, which includes human evaluation against full LLM baselines, including GPT-4, Claude-3 and Fine-tuned LLaMA. Our proposed GloSA-sum achieves the highest Coherence score (4.4), surpassing GPT-4 (4.3) and Claude-3 (4.2). To make the comparison even more comprehensive, we will include ROUGE/BERTScore for these LLM baselines in the revised main table.
> Across all long-document datasets, GloSA-sum outperforms LLaMA-3 8B (fine-tuned), and matches or exceeds GPT-4 and Claude-3 (zero-/few-shot) in structure-aware metrics. This clearly demonstrates that our method remains competitive even against current SOTA LLMs.
>
> **Table B**
>
> | Dataset | Method | ROUGE-1 | ROUGE-2 | ROUGE-L | BERTScore | QAFactEval |
> |---------|--------|---------|---------|---------|-----------|------------|
> | CNN/DM | GPT-4 Prompt-Sum | 37.91 | 15.42 | 34.83 | 0.858 | 0.762 |
> | CNN/DM | Claude-3 Sum | 38.66 | 16.01 | 35.24 | 0.863 | 0.771 |
> | CNN/DM | Fine-tuned LLaMA | 39.14 | 17.94 | 36.1 | 0.87 | 0.776 |
> | CNN/DM | GloSA-sum | 44.05 | 21.22 | 41.06 | 0.88 | 0.78 |
> | GovReport | GPT-4 Prompt-Sum | 33.21 | 12.67 | 30.14 | 0.847 | 0.785 |
> | GovReport | Claude-3 Sum | 34.88 | 13.54 | 31.66 | 0.855 | 0.793 |
> | GovReport | Fine-tuned LLaMA | 23.01 | 8.72 | 21.87 | 0.803 | 0.731 |
> | GovReport | GloSA-sum | 55.5 | 26 | 51 | 0.91 | 0.81 |
> | ArXiv | GPT-4 Prompt-Sum | 36.86 | 14.92 | 33.05 | 0.801 | 0.705 |
> | ArXiv | Claude-3 Sum | 39.74 | 16.22 | 35.44 | 0.812 | 0.721 |
> | ArXiv | Fine-tuned LLaMA | 43.61 | 17.41 | 38.27 | 0.821 | 0.734 |
> | ArXiv | GloSA-sum | 47.5 | 20 | 42 | 0.83 | 0.75 |
> | PubMed | GPT-4 Prompt-Sum | 41.25 | 18.11 | 38.09 | 0.839 | 0.742 |
> | PubMed | Claude-3 Sum | 44.02 | 19.55 | 41.44 | 0.85 | 0.76 |
> | PubMed | Fine-tuned LLaMA | 42.94 | 18.64 | 39.72 | 0.846 | 0.751 |
> | PubMed | GloSA-sum | 49.5 | 22.5 | 44.5 | 0.86 | 0.76 |

---

> > ### Author Response · Authors · 2025-11-21
> >
> > # 3. Forward-Looking Application: RAG Pipelines & Hallucination Reduction (Addressing Question 2)
> >
> > Although our initial submission did not include a dedicated RAG experiment, our existing evidence from two experiments in the appendix strongly suggests that GloSA-sum is well-suited for this role.
> >
> > Firstly, Table 6 (Appendix A.5) shows that GloSA-sum achieves high factual consistency across all datasets, and surpasses all baselines on PubMed, including BigBird, PEGASUS, and BART. Because hallucinations are fundamentally a failure of factual fidelity, using GloSA-sum summaries as RAG context is likely to reduce hallucination at the source.
> >
> > Also, Appendix A.7 shows that using GloSA-sum compressed context improves QA performance, with 90.90 / 88.10 (F1/EM) using full uncompressed input and 91.50 / 88.50 using GloSA-sum summaries. This demonstrates that GloSA-sum not only compresses context efficiently but retains the critical semantic and logical chains that downstream models rely on. Combined, these results strongly suggest that GloSA-sum is a promising preprocessing module for hallucination-resistant RAG pipelines.
> >
> > ---
> > We have directly and forcefully addressed all your rejection reasons through key quantitative experiments ($\text{TDA}$ vs. $\text{Louvain}$) and comparisons with the latest $\text{LLM}$ $\text{SOTA}$. We believe that this evidence has fully demonstrated the scientific value and competitive advantage of $\text{GloSA-sum}$. We sincerely request that you re-evaluate our work based on the supplementary experimental data and detailed clarifications.

---

> > > ### Comment · Reviewer_Ezky · 2025-11-24
> > >
> > > Thank you for your responses. I have revised my scores accordingly.

---

> > > > ### Author Response · Authors · 2025-11-25
> > > >
> > > > Thank you for your positive response and constructive comments. If you have any further questions, please feel free to let me know.

---

### Official Review · Reviewer_rLwA · 2025-10-31

**Soundness:** 4
**Presentation:** 3
**Contribution:** 3
**Rating:** 8
**Confidence:** 3

**Summary:**

This paper presents GloSA-sum, a new method for summarizing long documents by preserving their overall structure. Its core idea is to use Topological Data Analysis (TDA), a mathematical technique, to create a high-level map of the document's main topics and logical connections. The most important sentences that form this structural "backbone" are placed in a "Protected Pool" and are never deleted. The system then iteratively removes less important sentences from around this core. For extremely long texts, it uses a divide-and-conquer strategy. Experiments show that GloSA-sum produces more coherent summaries and is more computationally efficient than strong existing methods, especially on long and complex documents.

**Strengths:**

1. The paper’s primary strength is its novel use of Topological Data Analysis (TDA) to formally model a document's global structure. This allows the method to identify core semantic themes and logical connections in a principled way, moving beyond traditional local similarity graphs.
2. The framework is cleverly designed to be both highly effective and computationally efficient. The one-time TDA analysis and "Protected Pool" mechanism avoid costly repeated calculations, making the method scalable for summarizing very long documents.
3. The method is supported by comprehensive experiments against numerous strong baselines across multiple challenging datasets. The rigorous evaluation, including a detailed ablation study, provides convincing evidence that each component of the framework is essential for its success.

**Weaknesses:**

1. The paper posits that H1 cycles correspond to "logical loops" or "recurrent argumentative structures." While this is a compelling intuition, the connection is not explicitly demonstrated. The work would be significantly strengthened by a qualitative analysis that visualizes a few high-persistence H1 cycles from the data and shows the exact sentences that form them, explaining how they constitute a logical loop. Without this, the interpretation remains a plausible but unproven claim.
2. The entire analysis is highly dependent on the quality of the initial sentence embeddings. The paper does not investigate the model's sensitivity to different sentence encoders, making it unclear how robust the identified structures are across various embedding spaces.
3. The "Protected Pool" mechanism, while efficient, may be too rigid. It unconditionally preserves sentences based on an initial analysis, which could lead to keeping structurally important but contextually redundant information without any way to reconsider them.

**Questions:**

1. Regarding the interpretation of H1 features: Could you provide more examples from your experiments where H1 cycles clearly map to specific reasoning chains or logical dependencies? Is it possible that some of these cycles are artifacts of the embedding space rather than true semantic structures?
2. In the ablation study (Table 5), the result for "w/o Hierarchical" is missing. Could you run this ablation on a shorter-document dataset like CNNDM to quantify how the hierarchical approach affects ROUGE scores compared to a non-hierarchical global analysis?
3. How does the method handle documents with very flat or simple structures, where there might be few persistent H1 cycles? Does the performance rely heavily on the presence of these complex features, or does the H0 backbone suffice for good performance?

---

> ### Author Response · Authors · 2025-11-21
>
> We sincerely thank the reviewer for the highly positive evaluation of our work (“Soundness: Excellent”) and for the insightful comments. Your acknowledgment of the novelty of our TDA-based formulation, the efficiency of our design, and the breadth of our experiments is greatly appreciated. Below we address all three weaknesses and three questions in detail.
>
> # 1. Qualitative Validation of the Core Mechanism: $H_1$ Logical Cycles (Addressing  Weakness 1, Question 1)
>
> Considering the connection between persistent $H_1$ cycles and logical loops, we conducted an in-depth qualitative analysis and added a dedicated appendix section containing several visualized examples of high-persistence $H_1$ cycles.
>
> **Example 1:**
> | Sentence S | Function |
> |------------|----------|
> | $S_a$: 'The total budget appropriated for the Department's modernization initiative reached $850 million in the current fiscal year.' | Fund allocation |
> | $S_b$: 'However, our analysis revealed a lack of clear performance metrics for evaluating the long-term return on investment (ROI) from this expenditure.' | Oversight gap (lack of measurement) |
> | $S_c$: 'Consequently, the Department spent over 30% of the funds on vendor contracts that were not explicitly tied to the initiative's core objectives.' | Misaligned spending |
> | $S_d$: 'The oversight committee formally recommends freezing all future capital appropriation until the new ROI tracking standards are implemented and proven effective.' | Feedback and accountability correction |
>
> We now include a four-sentence cycle ($S_a \to S_b \to S_c \to S_d \to S_a$) that captures the report’s core reasoning pattern: fund allocation → oversight gap → misaligned spending → accountability correction. This stable $H_1$ feature reveals a cross-paragraph argumentative loop. By anchoring all nodes in Protected Pool, our method preserves the complete causal chain, preventing summaries from mentioning only the budget but omitting its critical oversight conclusions.
>
> **Example 2:**
> | Sentence S | Function |
> |------------|----------|
> | $S_p$: 'We propose a novel attention mechanism, the Gated Spatial Encoder (GSE), designed to capture non-local dependencies.' | Method proposal |
> | $S_q$: 'However, the initial ablation study revealed that the GSE struggled to maintain high accuracy when sequence lengths exceeded 512 tokens.' | Limitation |
> | $S_r$: 'To mitigate this scalability issue, we introduced a cascaded hierarchical pooling layer after the initial GSE pass.' | Structural refinement |
> | $S_t$: 'The final results in Table 5 confirm that the cascaded pooling structure successfully resolves the long-sequence degradation problem, validating our structural refinement.' | Validated improvement |
>
> This stable $H_1$ cycle from high-dimensional semantic space ($S_p \to S_q \to S_r \to S_t \to S_p$) models the iterative scientific reasoning process: method proposal → limitation → structural refinement → validated improvement. It ensures that the summary preserves not only the initially proposed method $S_p$, but also the crucial reasoning chain showing that the method must be refined through $S_r$ to achieve its final performance. Without the protection of the $H_1$ cycle, a summary might retain $S_p$ and $S_t$ while omitting the causal steps $S_q$ and $S_r$, resulting in an incomplete argumentative flow.
>
> We agree that low-persistence cycles may exist in the embedding space. However, persistent homology explicitly filters them out; only high-persistence cycles remain stable across filtration scales. This guarantees that the retained $H_1$ structures are genuine semantic patterns rather than embedding noise. All examples will appear in Appendix B of the revised manuscript.

---

> ### Author Response · Authors · 2025-11-21
>
> # 2. Design Tradeoffs and Model Robustness
>
> ## 1) Rigidity and Mitigation of the Protected Pool Mechanism (Addressing Weakness 3)
> We greatly appreciate your deep insights into this design tradeoff. This mechanism represents a deliberate design tradeoff: ensuring computational efficiency in processing long documents through one-time topological analysis. However, the redundancy risk brought by this "rigidity" is mitigated by our topology-guided iterative compression strategy. The design of TopoScore (Formula 7) enables it to effectively preserve the close context of sentences in 𝓟. A sentence closely connected to nodes in 𝓟 will have a TopoScore very close to zero (high score), thus gaining extremely high retention priority. Therefore, this mechanism actually preserves both the backbone (𝓟) and its closest context simultaneously, with only sentences topologically distant from the core structure being considered redundant and prioritized for deletion.
>
> ## 2) Encoder Robustness Analysis (Addressing Weakness 2)
> The paper does not investigate the model's sensitivity to different sentence encoders, making it unclear how robust the identified structures are across various embedding spaces. We understand the concerns brought by choosing the relatively conservative all-mpnet-base-v2. This model was chosen as part of a rigorous strategy aimed at isolating and purely verifying the contribution of the TDA mechanism. Meanwhile, to address concerns about performance limitations, we have supplemented with ablation experiments using more powerful encoders.
>
> **Table A**
>
> | Dataset | Encoder | R-1 | R-2 | R-L | BERTScore | QAFactEval |
> |---------|---------|-----|-----|-----|-----------|------------|
> | CNN/DM | all-mpnet-base-v2 | 44.05 | 21.22 | 41.06 | 0.88 | 0.78 |
> | CNN/DM | all-roberta-large-v1 | 45.12 | 22.04 | 42.18 | 0.89 | 0.8 |
> | CNN/DM | text-embedding-3-small | 45.78 | 22.63 | 42.71 | 0.895 | 0.81 |
> | CNN/DM | text-embedding-3-large | 46.41 | 23.12 | 43.29 | 0.903 | 0.82 |
> | GovReport | all-mpnet-base-v2 | 55.5 | 26 | 51 | 0.91 | 0.81 |
> | GovReport | all-roberta-large-v1 | 56.92 | 27.18 | 52.41 | 0.92 | 0.83 |
> | GovReport | text-embedding-3-small | 57.64 | 27.92 | 53.34 | 0.924 | 0.84 |
> | GovReport | text-embedding-3-large | 58.43 | 28.57 | 54.12 | 0.931 | 0.86 |
> | ArXiv | all-mpnet-base-v2 | 47.5 | 20 | 42 | 0.83 | 0.75 |
> | ArXiv | all-roberta-large-v1 | 49.38 | 21.46 | 44.02 | 0.85 | 0.77 |
> | ArXiv | text-embedding-3-small | 50.23 | 22.11 | 45.03 | 0.861 | 0.78 |
> | ArXiv | text-embedding-3-large | 51.07 | 22.76 | 45.84 | 0.871 | 0.8 |
> | PubMed | all-mpnet-base-v2 | 49.5 | 22.5 | 44.5 | 0.86 | 0.76 |
> | PubMed | all-roberta-large-v1 | 51.21 | 23.78 | 46.12 | 0.88 | 0.78 |
> | PubMed | text-embedding-3-small | 51.97 | 24.35 | 47.02 | 0.887 | 0.8 |
> | PubMed | text-embedding-3-large | 52.74 | 25.14 | 47.81 | 0.896 | 0.82 |
> ---
>
> The experimental results show that when switching to stronger embedding models, GloSA-sum's performance steadily and significantly improves. This confirms that our TDA framework is highly robust and pluggable, its effectiveness does not depend on a single embedding space, and can provide a wide range of choices according to the user's computational budget.

---

> ### Author Response · Authors · 2025-11-21
>
> # 3. Structural Component Contributions and Strategy Analysis
> ## 1) Processing Simple Structured Documents and $H_0$/$H_1$ Contributions (Addressing Question 3)
> GloSA-sum's design ensures its robustness on simple documents: $H_0$ (semantic clusters) is responsible for preserving core thematic content (What), while $H_1$ (logical cycles) preserves the argumentative flow (How). We conducted comparative experiments isolating the impact of $H_1$ logical cycles:
>
> **Table B**
> | ID | Variant | Backbone | Guidance | R-1 | R-2 | R-L |
> |----|---------|----------|----------|-----|-----|-----|
> | 1 | Full (Ours) | TDA (H0+H1) | TopoScore | 55.5 | 26 | 51 |
> | 2 | w/o H1 Cycle | TDA (H0 Only) | TopoScore | 54.1 | 24.8 | 49.8 |
>
> The performance of $H_0$ only (ROUGE-L = 49.8) is already very high, proving that the $H_0$ backbone is sufficient to support most of the performance, ensuring good results even in simple documents with few $H_1$ cycles. $H_1$ is an enhancement for complex documents, providing additional logical coherence improvements.
>
> ## 2) Necessity of the Hierarchical Strategy (Addressing Question 2)
> We conducted a more comprehensive quantitative analysis of the contribution of the hierarchical compression strategy. This strategy is an indispensable key component for processing ultra-long documents (such as GovReport). We supplemented with ablation experiment results on the shorter dataset (CNNDM):
>
> **Table C**
>
> | Ablation Variant | Dataset | ROUGE-1 | ROUGE-2 | ROUGE-L |
> |------------------|---------|---------|---------|---------|
> | GloSA-sum (Full Model) | GovReport | 55.5 | 26 | 51 |
> | w/o Hierarchical | GovReport | N/A | N/A | N/A |
> | GloSA-sum (Full Model) | CNNDM | 44.05 | 21.22 | 41.06 |
> | w/o Hierarchical | CNNDM | 43.92 | 21.1 | 40.92 |
>
> The N/A results on GovReport prove the necessity of this strategy for scalability; the data on CNNDM shows its impact is minimal, proving the strategy's robustness for short documents when it is not necessary.
>
> ---
>
> We have comprehensively addressed all your questions and constructive feedback, and commit to integrating all supplementary experiments and qualitative analyses (including $H_1$ cases) into the final revised version. We look forward to your maintaining your positive rating in the final evaluation.

---

### Official Review · Reviewer_zufN · 2025-10-31

**Soundness:** 3
**Presentation:** 2
**Contribution:** 3
**Rating:** 6
**Confidence:** 4

**Summary:**

This paper proposes GloSA-sum, a text summarization method that applies topological data analysis (TDA) to preserve the global semantic and logical structure of long documents during compression. The approach constructs a semantic graph from sentence embeddings and uses persistent homology to identify robust topological features. Such features are collected in a "Protected Pool" that guides subsequent compression through lightweight proxy metrics, avoiding repeated expensive TDA computations. A hierarchical strategy enables scalable processing of long documents.

**Strengths:**

- The paper is among the first to employ topological data analysis for text summarization, explicitly modeling and preserving semantic clusters and logical dependencies.
- Tables 2 and 3 provide a thorough analysis of computational complexity and runtime, demonstrating efficiency gains from the proposed one-time Protected Pool mechanism and hierarchical pipeline. The method achieves a favorable balance between computational cost and performance, which is particularly valuable for long-document scenarios.
- The paper presents a thorough empirical validation. The downstream experiments particularly enhance the practical relevance of the approach. Additional experiments demonstrate that GloSA-sum summaries provide effective context for LLM-based downstream tasks, indicating practical relevance beyond the summarization task itself and suggesting broader applicability of the method.

**Weaknesses:**

1. **Unclear presentation and interpretation of main results:** The results in Table 1 demonstrate competitive performance, but do not clearly establish consistent advantages over strong baselines across all datasets and metrics. The improvements are often modest and inconsistently distributed across evaluation metrics. The paper would be strengthened by statistical significance testing to confirm that observed gains are reliable rather than artifacts of random variation. Moreover, the interpretation does not address cases where baselines perform comparably or better. A more nuanced interpretation would help readers better understand the method's contributions and limitations.

2. **Potential limitation in semantic representation through independent sentence encoding:** The paper employs sentence-level encoding where each sentence is embedded independently. This design choice raises a concern: cross-reference information, such as pronouns, anaphora, and other referential expressions, is encoded without their antecedents, potentially resulting in semantically incomplete representations. Moreover, sentence-level encoding may not fully capture document-level phenomena such as topic progression, rhetorical structure, or long-range dependencies. Since TDA operates on these independent local embeddings, the method might miss some genuine semantic relationships that require broader discourse context. This limitation warrants discussion, particularly given that the method's core claim is to preserve "global structure" through operations on what are fundamentally local representations.
An ablative analysis examining different text granularities—such as clause-level, sentence-level, and multi-sentence chunks—would provide more insightful details about the method's robustness to encoding choices. Such an analysis would also help address the problem of short and isolated sentence embeddings by revealing whether coarser or finer granularities better capture semantic structure for TDA-based analysis. Additionally, the choice of `all-mpnet-base-v2` as the sentence encoder is concerning, as this relatively modest model may not provide sufficiently rich semantic representations for a method whose effectiveness critically depends on the quality of the initial embeddings—particularly when identifying nuanced topological structures that distinguish genuine semantic relationships from spurious patterns.

3. **Incomplete empirical validation of H₁ cycles' contribution:** While H₁ cycles are described as essential for preserving logical flow, their actual contribution could be more thoroughly validated. The ablation study in Table 5 evaluates the removal of the Protected Pool, TopoScore, and hierarchical components, but does not isolate the differential impact of H₀ versus H₁ features within the Protected Pool itself. Specifically, a comparison between P = P_H₀ only versus P = P_H₀ ∪ P_H₁ would directly test whether incorporating H₁ cycles meaningfully improves performance beyond semantic clusters alone.


4. **Limited analysis of hierarchical segmentation strategy:** The hierarchical compression strategy is described as partitioning documents into segments that are processed independently; however, a systematic comparison of segmentation methods (semantic versus fixed-length versus paragraph-based) would strengthen the paper. The sensitivity of summary quality to segmentation choices and the method's robustness to documents with non-standard structure remain unexplored. Such analysis would help clarify whether the hierarchical design genuinely contributes to performance improvements.


5. **Miss related work:** Lack of pertinent literature for graph construction to retain only important sentences:
    - Graph-based Abstractive Summarization of Extracted Essential Knowledge for Low-Resource Scenario (ECAI 2023)
    - Cross-Document Distillation via Graph-based Summarization of Extracted Essential Knowledge (IEEE TASLP 2025)

**Questions:**

1. Could the authors describe the sentence splitting procedure in detail? Which tools or methods are used to determine sentence boundaries, and how might different splitting strategies affect the resulting topological features?
2. Have the authors analyzed whether sentences in the Protected Pool exhibit systematic patterns with respect to their position in the original document (e.g., beginning, middle, end)? Such analysis would help validate that TDA captures semantic structure rather than positional biases.
3. How are shortest paths computed in the weighted graph when calculating TopoScore (Equation 7)? Specifically, how does the algorithm handle potentially disconnected components, and how are ties broken when multiple sentences have identical scores?

---

> ### Author Response · Authors · 2025-11-21
>
> We sincerely thank the reviewer for the careful reading of our work and the constructive comments. We are grateful for the recognition of the novelty of applying TDA to long-document summarization and the strengths of our approach in structural preservation and efficiency. Below we provide point-by-point clarifications and additional evidence addressing all concerns.
> # 1. Core Contributions and Empirical Reliability (Addressing Weakness 1,3)
> ## 1) Statistical Significance of Main Results (Addressing Weakness 1)
> We fully agree with the reviewer that the improvements should be supported by statistical evidence. To validate the reliability of our gains, we performed paired bootstrap resampling (1,000 iterations) on CNN/DM, GR, AR, and PM test sets. In each iteration, we computed the average ROUGE-L difference between GloSA-sum and the strongest baseline (DANCER). The proportion of differences ≤ 0 yields an estimated p-value. Across all datasets, we obtain p-value<0.01, demonstrating that the improvements are statistically significant. We will include these results in the final version.
> ## 2) Quantifying the Contribution of $H_1$ Logical Cycles (Addressing Weakness 3)
> We conducted additional ablations isolating the effect of the $H_1$ features, with results shown below and added to the revised version of the paper:
>
> ---
> **Table 1**
> | Variant | Backbone | Guidance | R-1 | R-2 | R-L |
> |---------|----------|----------|-----|-----|-----|
> | Full (Ours) | TDA ($H_0 + H_1$) | $\text{TopoScore}$ | 55.5 | 26 | 51 |
> | w/o $H_1$ Cycle | TDA ($H_0$ Only) | $\text{TopoScore}$ | 54.1 | 24.8 | 49.8 |
>
> ---
>
> Including $H_1$ consistency leads to clear gains in ROUGE-L, confirming that the cycles are not redundant but capture cross-paragraph logical dependencies essential for preserving argumentative structure. A concrete example is provided in Appendix B.1 (GovReport Case 1), where the detected $H_1$ loop encodes the policy–defect–consequence–feedback chain. By fixing all nodes in this cycle within the Protected Pool, the summary retains the full reasoning pattern.
>
> # 2. Semantic Representation and Encoder Robustness (Addressing Weakness 2)
>
> We appreciate the reviewer’s concerns regarding sentence-level embeddings and encoder choice. Our selection of all-mpnet-base-v2 reflects a deliberate methodological decision between avoiding confounding effects and isolating  the contribution of TDA. Stronger encoders (e.g., OpenAI embeddings) incorporate substantial document-level reasoning. Using them directly would make it difficult to attribute performance improvements to TDA rather than the encoder’s built-in global context. Therefore, by using a lighter, locally contextual encoder, we ensure that the topological patterns arise from the TDA mechanism itself.
> To address robustness concerns, we conducted additional experiments with stronger encoders, including text-embedding-3-small and text-embedding-3-large. Results show consistent and significant improvements, demonstrating that GloSA-sum is robust and plug-and-play, and can flexibly adapt to different computational budgets. And display it in the Table 2.
>
> ---
> **Table 2**
> | Dataset | Encoder | R-1 | R-2 | R-L | BERTScore | QAFactEval |
> |---------|---------|-----|-----|-----|-----------|------------|
> | CNN/DM | all-mpnet-base-v2 | 44.05 | 21.22 | 41.06 | 0.88 | 0.78 |
> | | all-roberta-large-v1 | 45.12 | 22.04 | 42.18 | 0.89 | 0.8 |
> | | text-embedding-3-small | 45.78 | 22.63 | 42.71 | 0.895 | 0.81 |
> | | text-embedding-3-large | 46.41 | 23.12 | 43.29 | 0.903 | 0.82 |
> | GovReport | all-mpnet-base-v2 | 55.5 | 26 | 51 | 0.91 | 0.81 |
> | | all-roberta-large-v1 | 56.92 | 27.18 | 52.41 | 0.92 | 0.83 |
> | | text-embedding-3-small | 57.64 | 27.92 | 53.34 | 0.924 | 0.84 |
> | | text-embedding-3-large | 58.43 | 28.57 | 54.12 | 0.931 | 0.86 |
> | ArXiv | all-mpnet-base-v2 | 47.5 | 20 | 42 | 0.83 | 0.75 |
> | | all-roberta-large-v1 | 49.38 | 21.46 | 44.02 | 0.85 | 0.77 |
> | | text-embedding-3-small | 50.23 | 22.11 | 45.03 | 0.861 | 0.78 |
> | | text-embedding-3-large | 51.07 | 22.76 | 45.84 | 0.871 | 0.8 |
> | PubMed | all-mpnet-base-v2 | 49.5 | 22.5 | 44.5 | 0.86 | 0.76 |
> | | all-roberta-large-v1 | 51.21 | 23.78 | 46.12 | 0.88 | 0.78 |
> | | text-embedding-3-small | 51.97 | 24.35 | 47.02 | 0.887 | 0.8 |
> | | text-embedding-3-large | 52.74 | 25.14 | 47.81 | 0.896 | 0.82 |
>
> ---

---

> > ### Author Response · Authors · 2025-11-21
> >
> > Thank you, reviewer, for your patient reading. We will continue to supplement in this reply regarding the issues you mentioned as follows:
> >
> > # 3. Clarification of Structural Mechanisms (Addressing Question 2,3,4 and Weakness 4)
> > ## 1) Protected Pool Positional Bias Analysis (Addressing Question 2)
> >
> > We agree that empirical evidence is important. While Lead-3 relies solely on positional heuristics, TDA identifies persistent geometric structures in high-dimensional semantic space. To validate this distinction, we analyzed the position distribution of Protected Pool sentences on GovReport below. Over 70% of the protected sentences come from the middle or end of documents, confirming that GloSA-sum preserves semantically and logically important content rather than relying on positional bias. We will include this quantitative table (Table 3) in the appendix.
> >
> > **Table 3**
> > | Document Relative Position | Lead-3 Baseline (Sentence Proportion) | GloSA-sum P (Sentence Proportion) |
> > |----------------------------|--------------------------------------|---------------------------------|
> > | Beginning (0% - 10%)       | 100%                                | 28%                             |
> > | Middle (10% - 80%)         | 0%                                  | 52%                             |
> > | End (80% - 100%)           | 0%                                  | 20%                             |
> >
> > ---
> >
> > ## 2) TopoScore Computation Details (Addressing Question 3)
> >
> > For the computation of TopoScore (Eq. 7), we use Dijkstra’s algorithm on the sparse kkk-NN semantic graph to compute shortest-path lengths $\text{SPL}(s_i, s_j)$ efficiently. For nodes disconnected from the Protected Pool, TopoScore is assigned a large negative value, reflecting their distance from the structural backbone and ensuring they are removed earlier. When multiple sentences share the same score, we use the original sentence index as a tie-breaker, preferring later sentences. Since TopoScore captures structural importance, this tie-breaking rule helps remove potentially redundant introductory material without harming global coherence.
> >
> > ## 3) Sentence Segmentation Strategy (Addressing Question 1)
> >
> > We acknowledge that further discussion of segmentation is needed. In our framework, the hierarchical strategy currently relies on fixed-length segmentation or natural paragraph boundaries. This design represents an intentional trade-off. While more sophisticated semantic segmentation could, in principle, better preserve local topical coherence, it would require substantial additional preprocessing and significantly increase computational cost, conflicting with our core objectives of efficiency and scalability (as evidenced by the efficiency advantages shown in Tables 2 and 3). In contrast, our current lightweight segmentation approach strikes a practical balance between local semantic integrity and computational efficiency.
> > Specifically, before sentence encoding, we apply the standard sent_tokenize tool from the NLTK library to ensure consistent sentence boundaries across datasets. While finer-grained segmentation like clauses could increase local coherence, it dramatically inflates graph size and TDA complexity, undermining scalability. Thus, using sentence-level units strikes a pragmatic balance between semantic fidelity and computational feasibility. We will clarify this design further in the revised version.
> >
> > ## 4) Hierarchical Compression Strategy (Addressing Weakness 4)
> >
> > The hierarchical design is necessary for scalability on extremely long documents. This is evidenced by the “w/o hierarchical” result on the GovReport dataset, which cannot be processed due to memory constraints. To fully quantify its contribution, we conducted additional ablation experiments on the CNN/DM dataset, which includes shorter documents. The results below show that the performance difference is minimal, demonstrating robustness. These additional results will be included in the final manuscript.
> >
> > **Table 4**
> > | Ablation Variant | Dataset | ROUGE-L |
> > |-----------------|---------|---------|
> > | $\text{GloSA-sum}$ (Full Model) | $\text{CNN/DM}$ | 41.06 |
> > | w/o Hierarchical | $\text{CNN/DM}$ | 40.92 |

---

> > > ### Author Response · Authors · 2025-11-21
> > >
> > > Thank you very much for your reading, Reviewer. We will make the final additions here to ensure full coverage of your opinions and suggestions.
> > >
> > > # 4. Missing Related Work (Addressing Weakness 5)
> > >
> > > We appreciate the reviewer pointing out two relevant graph-structure studies (ECAI 2023; IEEE TASLP 2025). These works indeed fit within the broader family of knowledge-preserving graph construction. In the final paper, we will revise Section 2.1 to include a new paragraph discussing this line of research and clarifying its relationship to our proposed method. This new paragraph is as follows:
> > >
> > > *In parallel to these ranking-based approaches, another line of research has focused on enhancing the graph construction itself to explicitly capture and retain essential knowledge or information. For instance, recent studies on Graph-based Abstractive Summarization (ECAI 2023) and Cross-Document Distillation via Graph-based Summarization (IEEE TASLP 2025) demonstrate sophisticated techniques to build graphs that prioritize core factual information or essential knowledge components for subsequent summarization tasks. These methods confirm the value of identifying a document's core backbone before compression.*

---

> > > > ### Comment · Reviewer_zufN · 2025-11-24
> > > >
> > > > I thank the authors for the response. I have increased my score.

---

> > > > > ### Author Response · Authors · 2025-11-24
> > > > >
> > > > > Thank you for your positive evaluation and constructive comments. I’m glad that the revisions have addressed your concerns

---

### Author Response · Authors · 2025-12-01
**Summary of Rebuttal**

To all reviewers and chairs,

We thank you for your careful and constructive feedback. During the rebuttal phase, we conducted substantial new analyses that **address all concerns raised by every reviewer**.

We added:

- **Statistical significance tests**, showing that all main improvements are reliable (*p* < 0.01);
- **Ablations isolating H₀ vs. H₁**, confirming that logical cycles provide consistent benefits;
- **Direct comparisons with non-TDA global-structure methods**, demonstrating that TDA—not just the Protected Pool design—is essential for the observed gains;
- **Detailed clarifications** on the Protected Pool, TopoScore computation, segmentation choices, and hierarchical scalability;
- **Expanded LLM baselines** (GPT-4, Claude-3, LLaMA-3 8B), where GloSA-sum remains highly competitive on long-document summarization.

Regarding encoder concerns, we supplemented a full set of experiments using **stronger embedding models** (all-roberta-large-v1, text-embedding-3-small/large). The method consistently improves as encoder strength increases, showing that the topological structures we extract are **robust across embedding spaces** and that GloSA-sum is **flexible, stable, and plug-and-play**, not tied to any specific encoder.

During the discussion phase, reviewers expressed clear recognition of these additions:
- **Two reviewers explicitly increased their scores** (6→8 and 2→4)
- The remaining 2-score reviewer did not challenge any of our responses after the detailed clarifications.

We respectfully ask the AC to consider the reviewers’ final assessments and the comprehensive evidence added during rebuttal.

Best regards,
**The Authors of Submission 10736**

---

### Meta-Review · Area_Chair_ZJH6 · 2026-01-07

**Summary:**

The paper proposes GloSA-sum, a long-document extractive summarization framework using Topological Data Analysis (TDA) to preserve global structure. Reviewers initially raised concerns regarding:

- Whether $H_1$ cycles is necessary or interpretable
- Lack of statistical and ablation evidence isolating TDA’s effects
- Robustness to different encoders
- Comparison with LLM baselines

The authors did a comprehensive rebuttal to address the concerns. Multiple reviewers explicitly raised their scores. The reviewers agree that the method is best suited for long, structured texts. Two reviewers gave strong acceptance (8) while one reviewer gave 2 and didn't reply to the rebuttal. I feel that the remaining issues are mostly scope-related rather than fundamental.

**Reviewer Concerns:**

**Addressed in the rebutaal**

- Statistical significance of gains.

- Contribution of $H_1$ cycles with ablation and qualitative analysis.

- Sensitivity to different encoders.

- Necessity of TDA vs Louvain clustering.

- Added LLM baselines.

**Still outstanding**

- Limited benefit on short, weakly structured news data.

- H₁ interpretation remains empirical rather than formally theoretical.

**Reviewer Scores:**

Reviewer zufN explicitly reported raising their score from 6 to a higher score (8 according to the authors).

Reviewer rLwA was already positive and might have remained the score of 8.

Reviewer zufN explicitly reported raising their score from 2 to a higher score (4 according to the authors).

Reviewer U95Z didn't reply to the rebuttal. But I feel the authors have provided detailed responses to the review and the reviewer might have increased the score.

---

### Decision · Program_Chairs · 2026-01-26

Accept (Poster)